



# Seasonal forecasting skill for the High Mountain Asia region in the Goddard Earth Observing System

Elias C. Massoud[1], Lauren Andrews[2], Rolf Reichle[2], Andrea Molod[2], Jongmin Park[3], Sophie Ruehr[1], Manuela Girotto[1]

[1]University of California Berkeley, Department of Environmental Science, Policy, and Management, Berkeley, CA, 94720 United States
[2]NASA Goddard Space Flight Center, Global Modeling & Assimilation Office, Greenbelt, MD, 20771 United States
[3]Goddard Earth Sciences Technology and Research (GESTAR II), University of Maryland, Baltimore, MD, 21250 United States

*Correspondence to*: Elias C. Massoud (eliasmassoud@berkeley.edu)

**Abstract.** Seasonal variability of the global hydrologic cycle directly impacts human activities, including hazard assessment and mitigation, agricultural decisions, and water resources management. This is particularly true across the High Mountain Asia (HMA) region, where water resource needs change depending on the seasonality and intensity of the hydrologic cycle. Forecasting the atmospheric states and surface conditions, including hydrometeorological relevant variables, at subseasonal-to-seasonal (S2S) lead times of weeks-to-months is an area of active research and development. NASA's Goddard Earth Observing System (GEOS) S2S prediction system has been developed with this research goal in mind. Here, we benchmark the forecast skill of GEOS-S2S (version 2) seasonal hydrometeorological forecasts in the HMA region, including a portion of the Indian Subcontinent, at 1-, 2-, and 3-month lead times during the retrospective forecast period, 1981-2016. To assess forecast skill, we evaluate 2-m air temperature, total precipitation, fractional snow cover, snow water equivalent, surface soil moisture, and terrestrial water storage forecasts against MERRA-2 and independent reanalysis, satellite observations, and data fusion products. Anomaly correlation is highest when the forecasts are evaluated against MERRA-2 and especially in variables with long memory in the climate system, possibly due to similar initial conditions and model architecture used in GEOS-S2S and MERRA-2. When compared to MERRA-2, results for the 1-month forecast skill ranges from anomaly correlation of $R_{anom}=0.18$ for precipitation to $R_{anom}=0.62$ for soil moisture. Anomaly correlations are persistently lower when forecasts are evaluated against independent observations; results for the 1-month forecast skill ranges from $R_{anom}=0.13$ for snow water equivalent to $R_{anom}=0.24$ for fractional snow cover. Hydrometeorological forecast skill is dependent on the forecast lead time, the memory of the variable within the physical system, and the validation dataset used. Overall, these results benchmark the GEOS-S2S system's ability to forecast HMA hydrometeorology on the seasonal timescale.

## 1 Introduction

Skillful prediction of hydrometeorological conditions at seasonal timescales depends on a range of factors, including the representation of land and ocean initial conditions (Dirmeyer et al., 2018; Mariotti et al., 2018), a model's ability to capture





large scale atmospheric processes (Gibson et al., 2020), a model's representation of climate mode variability (e.g., Waliser et al., 2006, 2009; Shukla et al., 2018), and the chosen perturbation and ensemble scheme (Scaife et al., 2014). Seasonal forecasting differs from numerical weather prediction, where skill largely depends on accurate representation of atmospheric

initial conditions (Pielke Sr. et al., 1999). The need to understand the processes driving seasonal prediction skill and the complexity of the systems needed to accurately represent said processes has driven extensive research (e.g.,    Merryfield et al., 2020; White et al., 2021). However, further improvements in seasonal forecasting skill, particularly of societally relevant variables, are sought because accurate subseasonal-to-seasonal (S2S) forecasts are useful for advance planning in various sectors, such as energy, water resources, agriculture, and disaster mitigation (National Academies Press, 2016).

S2S hydrometeorological forecasts can be particularly valuable in heavily populated regions, including High Mountain Asia (HMA) and the Indian Subcontinent, that experience substantial inter- and intra- annual variability in water resources. HMA has been dubbed one of the main 'water towers' of the Earth (Immerzeel et al., 2020) and has been hypothesized to influence global weather patterns through its impact on teleconnections (Nash et al., 2021). Seasonal forecasting systems, such as the

Goddard Earth Observing System S2S prediction system (GEOS-S2S), can skillfully capture large-scale atmospheric patterns and teleconnections (Gibson et al., 2020; Lim et al., 2021), including those impacting the HMA region. Ding and Wang (2007) and Lim (2015) demonstrated the importance of the Eurasian teleconnection in driving the planetary-scale Rossby-wave propagation that causes the intraseasonal variability over central Asia and the northern part of India. While other studies investigated climate variations over HMA by the impact of the North Atlantic Oscillation (Li et al., 2005,

2008), Indian Ocean Dipole and El Niño Southern Oscillation (Stueker et al., 2017; Sang et al., 2019; Power et al., 2021; Meena et al., 2022), the Central Indian Ocean mode (Zhou et al., 2017), and the boreal summer intraseasonal oscillation (Jiang et al., 2004; Hatsuzuka and Fujinami 2017).

The proper representation in S2S forecasting systems of the impact of large-scale teleconnections on hydrometeorological

conditions is complicated by local characteristics, causing S2S forecasts to degrade at local scales. This complication impacts the ability of seasonal forecasting systems to make accurate, high-resolution predictions on S2S timescales. For example, the northward propagation of the boreal summer intraseasonal oscillation originates in the northern Indian Ocean and tends to dissipate near the foot of Himalayas, and high humidity along the southern slope of the Himalayas and Tibetan Plateau leads to enhanced precipitation events (Jiang et al., 2004; Hatsuzuka and Fujinami 2017). It is, however, extremely

difficult to pin-point specific locations where this process ultimately occurs. Therefore, to gain a better understanding and make better predictions of how the Earth system behaves at regional scales, such as for the HMA region, further research is warranted.

Numerous investigations have examined the impacts of climate and weather in the HMA region, including air temperature

(Su et al., 2013; Dars et al., 2020), precipitation (Su et al., 2013; Ghatak et al., 2018; Liu and Margulis 2019; Christensen et



al., 2019; Dars et al., 2020; Stanley et al., 2020), terrestrial water storage and the overall water budget (Loomis et al., 2019a; Yoon et al., 2019), groundwater storage (Xiang et al., 2016; Wang et al., 2021), snow (Liu et al., 2021a; Liu and Margulis 2019; Margulis et al., 2019), glaciers (Shugar et al., 2020; Maurer et al., 2020; Batbaatar et al., 2021), atmospheric river storms (Nash et al., 2021), hydropower (Mishra et al., 2020), and landslides (Bekaert et al., 2020; Stanley et al., 2020). There 70  are many communities of scientists in the US, Europe, or China working on the HMA region (e.g., Arendt et al., 2017), investigating how climate is changing in HMA and what drives these changes.

More broadly, studies such as Vitart and Robertson (2018), de Andrade et al., (2019), and Robertson et al., (2020), have investigated and introduced the benefits of S2S forecasting for global climate and weather extremes. These types of studies 75  can be used to deduce the skill of seasonal forecasts for the HMA region. For instance, these studies show how forecasting a variable like precipitation in the HMA region can be difficult, with forecasts being acceptable out to week 1, but starting to degrade for forecasts in weeks 2-4. Furthermore, there have been studies that investigate the skill of seasonal forecasts specifically for regions including or close to HMA. For example, Deorias et al., (2021) compared the prediction of the Indian monsoon in different S2S models; Hsu et al., (2021) investigated simulations of the East Asian winter monsoon on S2S time 80  scales; Gerlitz et al., (2020) applied climate informed seasonal forecasts of water availability in Central Asia; and Zhou et al., (2021) developed a hydrological monitoring and S2S forecasting system for south and Southeast Asian river basins. Many of these studies utilized S2S prediction systems, but there is a need to evaluate seasonal forecasting for hydrometeorological variables in the HMA region. Our study examines the skill of seasonal forecasting for the HMA region using the GEOS-S2S forecasting system.


The Global Modeling and Assimilation Office utilizes the GEOS-S2S forecasting system, which initializes S2S forecasts each month using a weakly coupled atmosphere-ocean data assimilation system (Borovikov et al., 2018; Molod et al., 2020). Forecasts are provided to national and international multi-model prediction efforts, including the North American Multi-Model Ensemble (Kirtman et al., 2014). Recent GEOS-S2S system developments improved the representation of ocean 90  temperatures and heat transport (Molod et al., 2020) and the retrospective forecast of climate indices, including the El Niño Southern Oscillation, North Atlantic Oscillation, and the Madden-Julian Oscillation, particularly at 1-to-3-month lead times (Molod et al., 2020; Lim et al., 2021). These improvements should contribute to enhancements in global hydrometeorological forecast skill in GEOS-S2S.

In this study, we examine the ability of GEOS-S2S forecasts to accurately predict near-surface air temperature, total precipitation, fractional snow cover area, snow water equivalent, surface soil moisture, and terrestrial water storage across the HMA region and a large portion of the Indian subcontinent at 1-, 2-, and 3-month lead times. These variables are directly relevant to the accurate prediction of water resources and processes critical to local populations. Evaluation and improvement of hydrometeorological forecast lead time can improve warning systems for natural hazards such as flooding or



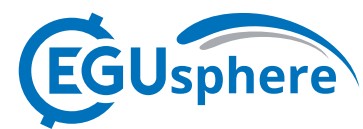

landslides and provide critical information for agricultural purposes (Bekaert et al., 2020; Stanley et al., 2020). However, the complexities of the relationships among variables as well as the regional topography within HMA make seasonal forecasting for this area challenging. Some of these variables, such as temperature or precipitation, are more difficult to accurately forecast at the S2S time scales compared to other variables because of their fast nature and low memory in the physical system. It is hypothesized there is higher forecast skill for the variables with longer temporal memory in the physical system,

such as snow, soil moisture, or terrestrial water storage. Therefore, proper initialization of these variables can allow for longer-lasting skill in the S2S forecasting system.

The first objective of this work is to provide a benchmark of GEOS-S2S hydrometeorological forecast skill for the HMA region and across a large portion of the Indian Subcontinent. A second objective of the analysis is to determine potential

areas of improvements in model initialization or more realistic representation in the model architecture, which can help enhance the forecast accuracy in future GEOS-S2S versions and extend the skillful forecast window of variables in the HMA region. The paper is organized as follows: Section 2 introduces the datasets and methods used in this study, Section 3 reports the results of the evaluation, Section 4 offers a discussion on the main findings, and Section 5 concludes with a summary of the paper.

## 115  2 Data and Methods

### 2.1 Region of Focus

Here, we refer to "HMA" as the domain shown in Figure 1 and covering parts of China, Afghanistan, Pakistan, Nepal, Bhutan, India, Bangladesh, Myanmar, Kazakhstan, Uzbekistan, Kyrgyzstan, and Tajikistan and stretches across five mountain ranges, including the Himalayas, Inner Tibetan Plateau, Karakoram, and Hindu Kush. These mountains funnel

fresh water into major river basins that support about 1.5 billion people, providing drinking water, irrigation, and hydropower (Immerzeel et al., 2020), including the Tarim, Indus, Yangtze, and Ganges/Brahmaputra basins. HMA has one of the highest concentrations of snow and glacier ice outside of the polar regions, making it an extremely important region to study and evaluate S2S forecasting of hydrometeorological variables.

### 2.2 GEOS-S2S Prediction System

We evaluate GEOS-S2S, version 2 (Molod et al., 2020); the GEOS-S2S forecasting system is an atmosphere-ocean general circulation model (AOGCM) and ocean data assimilation system. The AOGCM includes the GEOS atmospheric general circulation model (AGCM; Molod et al., 2015; Rienecker et al., 2008), the Catchment land surface model (Koster et al., 2000), Version 5 of the Modular Ocean Model developed by the Geophysical Fluid Dynamics Laboratory (Griffies, 2012), and Version 4.1 of the sea ice model developed by the Los Alamos National Laboratory (Hunke & Lipscomb, 2008). GEOS-

S2S forecasts are initialized using a precomputed atmospheric analysis and ocean data assimilation (Penny et al., 2013). The



system components are coupled using the Earth System Modeling Framework (Hill et al., 2004) and the Modeling Analysis and Prediction Layer interface layer (Suarez et al., 2007).

The GEOS-S2S forecasts are initialized from the weakly-coupled atmosphere-ocean data assimilation system using a 5-day
assimilation cycle. During the initial 5-day predictor segment, every 6 hours, the departure of model trajectory from observed ocean fields is determined and sea ice fraction is replaced with satellite-derived observations (Cavalieri et al., 1996). Following the predictor segment, the model is rewound and ocean analysis increments are applied during the first 18 hours of the 5-day corrector segment. During both segments, the atmosphere is nudged to a precomputed state and SST is strongly relaxed to MERRA-2 values to ensure that the ocean and atmosphere are as consistent as possible. A detailed
description is in Molod et al., (2020).

Forecasts are initialized at the end of the corrector segment. During the retrospective forecast period (1981-2016), forecasts are initialized using an unperturbed lagged scheme, with unperturbed forecasts initialized every 5 days during the last half of each month for four total ensemble members. During the forecast period, an additional 6 perturbed forecasts are initialized
on the last forecast day of each month. All forecasts are 9 months in duration, but here, we focus on retrospective forecasts with 1-, 2-, and 3-month lead times. Retrospective forecasts are completed to provide a model climatology for use in probabilistic forecasting and provide a long period for forecast validation (Molod et al., 2020). GEOS-S2S forecasts have been used and evaluated in studies related to the Madden-Julian Oscillation (Lim et al., 2021), sea surface salinity and its impact on the El Niño Southern Oscillation (Hackert et al., 2020), the impact of volcano eruptions on surface temperatures
and precipitation (Aquila et al., 2021), and others.

The hydrometeorological variables of interest were obtained from the "vis2d" or the "surf" collections of the GEOS-S2S archive, and include: 2-m air temperature (T2M from the "surf" collection), total precipitation (PRECTOT from the "vis2d" collection), snow cover area fraction (ASNOW from the "vis2d" collection and called fSCA for the remainder of this paper),
snow water equivalent (SNOMAS from the "vis2d" collection and called SWE for the remainder of this paper), soil moisture in the surface layer from 0-5 cm (WET1 from the "vis2d" collection and SM for the remainder of this paper), and terrestrial water storage (TWLAND from the "surf" collection and 'TWS' for the remainder of this paper). SM is calculated at each grid cell by multiplying the WET1 variable with porosity and then dividing by the density of water. For grid cells that are frozen or are covered in snow, the soil moisture value is masked out as a no-data-value grid cell, following the work of De
Lannoy and Reichle (2016). Simulated TWS includes soil moisture, snow, and the canopy interception reservoir, but not surface water (that is, lake and river water) or glaciers. Table 1 provides a list of these variables as represented in GEOS-S2S (Nakada et al., 2018), and the corresponding evaluation datasets, detailed in the following subsections.



## 2.3 Evaluation Datasets

The first product used here to evaluate the GEOS-S2S forecasts is the Modern-Era Retrospective analysis for Research and
Applications, version 2 (MERRA-2; Section 2.3.1; Gelaro et al., 2017). MERRA-2 and GEOS-S2S output includes many
compatible variables because the version of the GEOS AGCM in GEOS-S2S-2 is similar to the version used for the
production of the MERRA-2 reanalysis.

To further evaluate GEOS-S2S, we also use independent reanalysis and observational products (Table 1). To this end, for air
temperature we use the fifth-generation atmospheric reanalysis from the European Centre for Medium-Range Weather
Forecasts (ECMWF) reanalysis product (ERA5; Section 2.3.2; Hersbach et al., 2020). For precipitation, we use the Asian
Precipitation Highly Resolved Observed Data Integration Towards Evaluation product (APHRODITE; Section 2.3.3;
Yatagai et al., 2012). For snow cover, we use the Moderate Resolution Imaging Spectroradiometer (MODIS; Section 2.3.4;
Hall et al., 2002) remotely sensed product. For SWE, we use the HMA Snow Reanalysis product (HMA-SR; Section 2.3.5;
Margulis et al., 2019; Liu et al., 2021b). For soil moisture, we use the European Space Agency's Climate Change Initiative
data (ESA-CCI; Section 2.3.6; Dorigo et al., 2017). Lastly, for TWS, we use data from the NASA Gravity Recovery and
Climate Experiment satellite mission (GRACE; Section 2.3.7; Tapley et al., 2004). We utilize information from different
sources to make sure that evaluation results are not solely dependent on biases or uncertainties in a single reference product.

### 2.3.1 MERRA-2

MERRA-2 is the most recent NASA global atmospheric reanalysis product and is generated using the GEOS atmospheric
model and analysis (Gelaro et al. 2017). MERRA-2 output contains similar variables to GEOS-S2S and offers a rich product
to apply systematic evaluation of the model forecasts. We obtain information from MERRA-2 on all the variables of interest
listed in the previous section (Table 1) and for the same period (1981-2016; Bosilovich et al., 2016). The variables of interest
include: 2-m air temperature (T2M from the "Single-Level Diagnostics" collection; GMAO 2015a), total precipitation
(PRECTOT from the "Surface Flux Diagnostics" collection; GMAO 2015b), snow cover area fraction (FRSNO from the
"Land Surface Diagnostics" collection and called fSCA for the remainder of this paper; GMAO 2015c), snow water
equivalent (SNOMAS from the "Land Surface Diagnostics" collection and SWE for the remainder of this paper), soil
moisture in the surface layer from 0-5 cm (GWETTOP from the "Land Surface Diagnostics" collection and SM for the
remainder of this paper), and terrestrial water storage (TWLAND from the "Land Surface Diagnostics" collection and TWS
for the remainder of this paper). MERRA-2 uses observation-based precipitation data as forcing for the land surface
parameterization (Reichle et al., 2017). The precipitation forcing data derived from this approach is archived as the output
variable called PRECTOTCORR, which is available as part of the "Surface Flux Diagnostics"" collection (GMAO 2015b).
This variable is compared to the PRECTOT variable in the GEOS-S2S forecasts. Similar to GEOS-S2S, SM is calculated at
each grid cell by multiplying the GWETTOP variable with porosity and then dividing by the density of water.

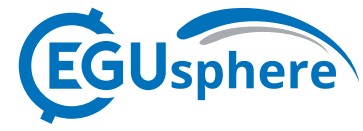

### 2.3.2 ERA5 2-m Air Temperature

For our study, we obtain information on T2M from ERA5. ERA5 covers the period from January 1950 to present and is available from the Copernicus Climate Change Service at ECMWF. ERA5 embodies a detailed record of the global atmosphere, land surface and ocean waves (Hersbach et al., 2020). The surface analysis in ERA5 ingests station observations of T2M where available and under suitable, warm-season conditions (De Rosnay et al., 2014). For times and locations where T2M observations are assimilated, the ERA5 T2M estimates are therefore closer to observations. In HMA, however, this is not necessarily the case, owing to the topographically complex terrain and generally colder conditions.

### 2.3.3 APHRODITE Precipitation

APHRODITE's gridded precipitation is a set of long-term, continental-scale, daily products that is based on a dense network of rain-gauge data for Asia. The data include information for many regions, including the Himalayas, South and Southeast Asia and mountainous areas in the Middle East from January 1951 until December 2015. We obtain information on the total precipitation from APHRODITE (version V1901; Yatagai et al., 2012) in our evaluation, which we utilize as the alternative observation for precipitation. The data are aggregated from daily to monthly time steps, and regridded from 0.05° to 0.5° resolution to match the model grid. There was no further quality control done on the data since this was already conducted by the data provider (Maeda et al., 2020).

### 2.3.4 MODIS Snow Cover Area

MODIS MOD10C1 Version 6 provides the daily (~10:30am local time) percentage of snow-covered land and cloud-covered land on the MODIS Climate Modeling Grid (posted at 0.05°; Hall and Riggs, 2016a). The MOD10C1 CMG dataset is generated from the Normalized Difference Snow Index snow cover of MOD10A1 (Hall and Riggs, 2016b) by mapping the 500 m MOD10A1 observation types (snow, snow-free land, cloud, etc.) to 0.05° bins. Snow and cloud cover percentages are derived by calculating the ratio of 500 m snow and cloud cover observations to the total number of 500 m land observations within each CMG grid cell. MOD10C1 also has basic quality assurance flags; data with a quality assurance flag other than 0, 1 or 2 was not used in this analysis. Daily data between February 2000 and December 2016 are averaged to create monthly mean snow cover percentages.

### 2.3.5 HMA-SR Snow Water Equivalent

The HMA-SR assimilates Landsat- and MODIS-derived fractional snow-covered area to derive seasonal snow water equivalent in HMA where in situ data are limited (Margulis et al., 2019; Liu et al., 2021b). The method is a probabilistic data assimilation version of a snow reconstruction approach, where SWE information is retrieved from the accumulation of melt events driven by energy forcings (i.e., downscaled global datasets for forcing a snow model) and observed snow cover area disappearance. The data product provides snow depth as well as SWE estimates from October 1999 to September 2017. The



data are aggregated from daily to monthly time steps, and regridded from 16 arc-seconds (~0.0044°) to 0.5° resolution to match the model grid. We used a non-seasonal snow mask to exclude grid cells with permanent snow and ice from the evaluation (Liu et al., 2021a).

### 2.3.6 ESA-CCI Soil Moisture

The ESA-CCI Programme on Global Monitoring of Essential Climate Variables was initiated in 2010 and produces an
updated soil moisture product every year (Dorigo et al., 2017; Gruber et al., 2019; Preimesberger et al., 2020). The ESA-CCI SM product comprises the three well-known active, passive, and combined microwave satellite soil moisture datasets from 1978 to 2020. In this study, information on soil moisture in the surface layer (0-5 cm) from ESA-CCI (version 6.1) is utilized in our evaluation. The ESA-CCI SM dataset is representative of the first few centimeters of soil (~0-5cm); however, an exact depth cannot be determined as multiple sensors are combined to create the data products. The depth depends on the available
sensors for each point in time and their characteristics (active/passive sensor, measurement frequency, etc). There are gaps in the original ESA-CCI data due to the quality control applied during post-processing, such as areas that are masked out for ice and snow (different time steps will have different masks applied). Furthermore, the product quality changes over time with the number and type of sensors integrated into the product, with more recent retrievals being generally of higher quality.

### 2.3.7 GRACE Terrestrial Water Storage

From 2002 to 2017, NASA's twin Gravity Recovery and Climate Experiment (GRACE) satellites monitored large-scale water storage changes all over the globe (Tapley et al., 2004; Rodell et al., 2009; Famiglietti et al., 2011; Massoud et al., 2018, 2021, 2022). GRACE provided estimates of global mass change at monthly resolution and at a relatively coarse spatial resolution (~300 km). Information on TWS from GRACE captures the dynamic signature of all water sources on the ground, such as surface reservoirs, lakes, rivers, glaciers, canopy water, soil moisture, snow, and groundwater. For our study,
we utilize the GRACE mascon product (Loomis et al., 2019b), available from April 2002 through June 2017.

### 2.4 Forecast Evaluation

For the evaluation of all variables with MERRA-2, we use monthly averaged forecasts from 1981-2016. For the verification with the reference data products, we also utilize monthly averaged data, yet the time periods differ for each of the reference data products, depending on the availability and quality of the reference data. The length of record of each data product used
is indicated in Table 1.

### 2.4.1 Calculating Climatologies and Anomalies

S2S forecasts can be assessed based on anomaly skill, i.e., the departure from expected normal conditions for a particular month (Kirtman et al., 2014). For this study, we remove the forecast climatology (i.e., the long-term mean value for each calendar month throughout the length of the available data record) for all analyzed variables. For an example of how this is





estimated, consider the calculation of the 1-month lead anomaly that is initialized in January and has a forecast in February. For this, we take all the 1-month lead forecasts for February between 1981-2016 and calculate their mean. This climatology will be subtracted from the forecast of February conditions that were initialized in January (i.e., 1-month lead), to determine the anomaly for that forecast. The same procedure is applied on the 2-month and 3-month forecasts to develop their respective anomalies. For the evaluation datasets, monthly climatologies were created using the time intervals defined in the previous sections and Table 1 and subtracted from each respective dataset.

### 2.4.2 Regridding and masking

All the data products listed above are spatially aggregated to a half degree resolution to match the grid size of the GEOS-S2S forecasts. For the case of higher-resolution data, such as MODIS (0.05 degree), the aggregation was done by computing the average across all the grid cells within each half degree grid cell in GEOS-S2S. For products with lower resolution, such as GRACE (1 degree posting), the data were re-gridded to half degree grids using bilinear interpolation to match the resolution of S2S forecasts. Furthermore, all the data is aggregated to monthly averages to facilitate the temporal comparisons with the S2S forecasts. There are cases where grid cells were masked out for the analysis for reasons such as availability or quality of the data. The masked-out data were removed in the calculation of the evaluation metrics, which are described in the next section.

### 2.4.3 Evaluation Metrics

For evaluating the 1-, 2-, and 3-month forecasts from GEOS-S2S, we use the monthly anomalies from each data set (section 2.4.1) and estimate the unbiased Root-Mean-Square-Error (ubRMSE) as well as the anomaly correlation ($R_{anom}$) between the model forecasts and the reference data.

The ubRMSE score is calculated as the RMSE of the anomaly forecasts from all grid cells and at all timesteps, as follows:

$$ubRMSE = 1/\sqrt{n} * \sqrt{\sum_{x,y,t} \ [\ GEOS.S2S_{anom}(x,y,t) - Ref.Data_{anom}(x,y,t)\ ]^2} \ , \quad (1)$$

where $GEOS.S2S_{anom}$ is the forecast anomaly from the S2S system at location (x,y) and at time (t), $Ref.Data_{anom}$ represents the anomaly of the reference data product used for the evaluation, and $n$ represents the number of elements in the calculation, which is a product of the number of grid cells and the number of time steps. For evaluation estimates that include masked-out grid cells, n is reduced to represent the total number of elements that are accounted for in the calculation.

The $R_{anom}$ score is calculated as the correlation of the anomaly of the S2S forecasts with the anomaly of the verification data. This score is estimated using the 'corrcoef' function in MATLAB

(https://www.mathworks.com/help/matlab/ref/corrcoef.html; Press et al., 1992), which also provides upper and lower limits
that can be used for estimating the error bars around the correlation estimate. We report the error bars around the $R_{anom}$ score
by representing the interquartile range of the anomaly correlation from all the considered grid cells.

In this study, we also report on the ensemble spread of the GEOS-S2S forecasts. For estimating the ensemble spread for the
S2S forecasts, we calculate the standard deviation of the ensemble members from GEOS-S2S for each grid cell and at each
monthly time step. Since there are only 4 ensemble members at each time step and for each grid cell, the ensemble spread
can be rather noisy. Therefore, we estimate the ensemble spread as the mean of the long-term standard deviation of the
ensemble members at each grid cell. This helps with reducing the noise in the ensembles.

## 3 Results

In this section, we report the results of the evaluation, showing the skill of the GEOS-S2S hydrometeorological forecasts for
the HMA region. For reference, Table 2 lists the ubRMSE and the $R_{anom}$ for all variables when comparing the S2S forecasts
to the reanalysis (MERRA-2, Section 2.3) and the reference data products (Section 2.4). Further discussion of the results is
provided in Section 4.

### 3.1 Difference in Skill Among Variables and Forecast Lead Times

Table 2 and Figure 2 show the anomaly correlation for each variable and for each lead time considered, along with the error
bars for each anomaly correlation assessment. The results indicate that across all variables, the forecast skill at 1-month lead
is higher than at 2-month lead, which is higher than at 3-month lead. For example, for T2M, the 1-month forecast anomaly
correlation when compared to MERRA-2 is $R_{anom}$=0.24, for the 2-month forecast it is 0.13, and for the 3-month forecast it is
0.11. And when compared to ERA5, the 1-month forecast anomaly correlation for T2M is $R_{anom}$=0.19, for the 2-month
forecast it is 0.13, and for the 3-month forecast it is 0.10. Higher anomaly correlation for forecasts with shorter lead time is
shown regardless of which data product was used for the evaluation, which is aligned with other studies that evaluate S2S
forecasts (e.g., Deflorio et al., 2019; Molod et al., 2020).

When comparing the S2S forecasts to MERRA-2, the variables with longer memory in the physical climate system, such as
SM or TWS, have higher accuracy in the seasonal forecasting system compared to variables that represent more quickly
changing processes, such as T2M or PRECTOT (Figure 2a). When the reference data products are used in the evaluation
(Figure 2B), results show that there is little evidence that variables with longer memory have higher forecast accuracy, since
there is similar skill for forecasting most variables; the range of $R_{anom}$ for all variables in the verification results is 0.13 to
0.24.





Figure 3 shows the relative skill as a function of lead time for the six variables, where blue indicates higher and red lower relative skill. The relative skill refers to the ubRMSE of each variable at a given lead time, normalized by the average ubRMSE across all lead times. Figure 3 further illustrates the above finding that there is higher skill for forecasts with shorter lead times. For example, when comparing T2M forecasts against MERRA-2, the relative skill for forecasts with 1-month lead time is ~0.1, for 2-month lead time is close to 0, and for 3-month lead time it is about -0.1 (Figure 3A).

Furthermore, the difference in the colorbar bounds indicates that there is a larger range of skill for forecasts with different lead time when comparing GEOS-S2S with MERRA-2 (-0.2 to 0.2 in Figure 3A) than when comparing the forecasts to the reference data products (-0.05 to 0.05 in Figure 3B). This difference in the colorbar range between Figures 3A and 3B supports the finding that there is a larger drop off in skill with longer lead time when evaluating the GEOS-S2S forecasts against MERRA-2; when the evaluation is done against the reference data products, this drop off in skill is less pronounced.

**3.2 Annual cycles**

Figure 4 shows the annual cycle, averaged over HMA, of all data products considered in this study. For T2M (Figure 4A), the GEOS-S2S forecasts, MERRA-2, and ERA5 all have very similar annual cycles; this is persistent across lead times. The peak of the T2M annual cycle occurs during the summer months (June, July, August) reaching 290-295 K, and the low occurs during the winter months (December, January February) dropping to 273-275 K. GEOS-S2S PRECTOT forecasts

(Figure 4B) have a wet bias compared to the MERRA-2 and APHRODITE products across nearly the entire annual cycle. The peak of all products occurs in the summer months (JJA) reaching ~4-4.5 mm/day for the S2S forecasts and 3.5-4 mm/day for the evaluation products, and a low in the winter months (NDJF) dropping to 0.5-0.75 mm/day for the S2S forecasts and 0.25-0.5 mm/day for the evaluation products.

GEOS-S2S fSCA forecasts have more snow cover compared to the MERRA-2 and MODIS products, with a consistently higher mean and amplitude in the S2S forecasts (Figure 4C). As expected, the peak of all fSCA products occurs in the winter months (DJF) and the low in the summer months (JAS), however the magnitude and amplitude are different between the products. The peak fSCA in the S2S forecasts reaches ~0.23-0.25 and the low is about 0.05 for the S2S forecasts, and for the evaluation products the peak is only about 0.1-0.15 with lows that are less than 0.03. Furthermore, the annual cycle of

fSCA in MERRA-2 is consistently the lowest out of all the products. For SWE (Figure 4D), the GEOS-S2S forecasts have a different annual mean and amplitude for the various lead times and in comparison to the evaluation products. The peak SWE in the S2S forecasts occurs in the spring months (March and April), reaching a high of about 0.02 m for the 1-month lead forecasts, 0.025 m for the 2-month lead forecasts, and 0.03 m for the 3-month lead forecasts. For the HMA-SR product, the seasonality has a higher amplitude and magnitude, reaching a peak of ~0.03 m in the spring months. For MERRA-2, the

annual cycle of SWE is consistently lower compared to the other products, reaching a peak of ~0.005 m in February. All products show a minimum SWE of less than 0.005 m in the summer months.



For SM, the annual cycle of the GEOS-S2S forecasts is like that of the MERRA-2 product but is substantially different from the annual cycle of the ESA-CCI data (Figure 4E). The peak SM in the S2S forecasts occurs in the fall (~October) and

reaches ~0.25 m³/m³ and the low occurs in the spring (around May) and drops to about 0.12 m³/m³. This is similar in MERRA-2, with a peak of just over 0.2 m³/m³ that occurs in November and a low of just under 0.15 m³/m³ that occurs in May. For the ESA-CCI, the peak SM reaches ~0.27 m³/m³ and is observed in the summer months (JJAS) and the low drops to 0.17 m³/m³ and is observed in the early spring (March). Similarly, for TWS (Figure 4D), the annual cycle of the GEOS-S2S forecasts is like that of the MERRA-2 product but is substantially different from the annual cycles of the GRACE data.

The peak of the TWS anomaly in the S2S forecasts and in MERRA-2 occurs in the late summer (ASO) and reaches ~0.05 m and the low occurs in the spring (April and May) and drops to about -0.02 m. For GRACE, the peak TWS reaches a high of 0.05 m in the summer (JJA) and drops to a low of -0.05 m in the spring (March-April). Therefore, there is a 1–2-month temporal lag as well as a difference in the mean and amplitude of the annual cycles of the different products of SM and TWS between the various products considered.

**3.3 Error by Forecast Month**

The S2S forecast skill depends on various factors, such as the lead time or the variable of interest. Our results in Figures 5 and 6 show that skill also depends on the month that is forecasted. We observe this behavior in GEOS-S2S when compared to both MERRA-2 (Figure 5) and reference data products (Figure 6). Figures 5 and 6 show the area-averaged error based on the forecast month of interest for each variable. As an example, the three bars for the month of April include the 1-month, 2-

month, and 3-month forecasts, which are the forecasts initialized in March, February, and January respectively. These results can be suitable for those interested in understanding the errors of a forecast for a specific month, say April, using forecasts that were made 1, 2, or 3 months prior.

We find that GEOS-S2S forecasts of T2M have less skill in the winter season with ubRMSE greater than 2 K around

February and more skill in the summer season with ubRMSE of less than 1.5 K around August (Figures 5A and 6A). Errors in the precipitation forecasts are higher in the summer (July-August) compared to the winter months (December-January), with ubRMSE that is greater than 2 mm/day in the summer and less than 0.5 mm/day in the winter (Figures 5B and 6B).

For the snow variables, forecasts of fSCA have higher errors in the winter season (December-February) with ubRMSE close

to 0.1, and less error in the summer season (July-August) with ubRMSE of less than 0.01 (Figures 5C and 6C). For SWE, results are different when comparing the S2S forecasts with MERRA-2 and with the HMA-SR product. Figure 5D shows that when comparing the S2S forecasts of SWE to MERRA-2, there are higher errors in the spring (March-April) with ubRMSE of 1-1.5 cm and lower errors in the summer (August-September) with ubRMSE of less than 0.1 cm, with the forecast lead time impacting the amount of error. Yet, Figure 6D shows that when comparing the S2S forecasts of SWE to





the HMA-SR product, there are higher errors is in the summer months (July-August) with ubRMSE of 4 cm and lower errors in the fall (October-November) with ubRMSE close to 1 cm.

Errors in the SM forecasts are higher in the summer (July-August) compared to the winter months (February-April), with ubRMSE values up to 0.03 m$^3$/m$^3$ in the summer and as low as 0.01 m$^3$/m$^3$ in the winter and spring (Figures 5E and 6E), and

with the forecast lead time impacting the magnitude of error. For TWS forecasts, results are different when comparing the S2S forecasts with MERRA-2 and with the GRACE data. Figure 5F shows that when comparing the S2S forecasts of TWS to MERRA-2, there are higher errors in the summer (around August) with ubRMSE that is over 4 cm and lower errors in the winter (around February) with ubRMSE as low as 2 cm, with the forecast lead time impacting the magnitude of error. Yet, Figure 6F shows that when comparing the S2S forecasts of TWS to the GRACE data, there are higher errors in the spring

months (around April) with ubRMSE greater than 15 cm and lower errors in the winter (around February) with ubRMSE below 10 cm.

### 3.4 Spatial Patterns: Climatology, Ensemble Spread, and Forecast Error

This section focuses on the spatial aspect of the evaluation. Figures 7-12 show, for each variable, the GEOS-S2S ensemble mean climatology (1981-2016), the ensemble spread, the ubRMSE versus MERRA-2, and the ubRMSE versus the reference

data products. The top rows of these figures show the results for the 1-month forecasts, and the middle and bottom rows show the differences in the 2-month and 3-month forecasts with respect to the 1-month forecasts.

### 3.4.1 Evaluation of Temperature and Precipitation

As expected, T2M is generally higher at lower elevations, for example in India and Pakistan, and it is much cooler in the mountains and at higher elevation, for example in the Himalayas and the Inner Tibetan Plateau (Figure 7A). The ensemble

spread of T2M (Figure 7B) is low compared to the ubRMSE (Figure 7C and 7D), indicating that most ensemble members forecast similar T2M values. The ubRMSE is larger in regions where the spread is higher, indicating that the spatial patterns of the ensemble spread and ubRMSE are similar. ubRMSE relative to MERRA-2 and ERA5 show a similar magnitude of error throughout most of the domain, with ubRMSE values of up to ~3 K (Figure 7C and 7D). However, for the Inner Tibetan Plateau, there is more agreement with MERRA-2 (ubRMSE of ~2 K) than with the ERA5 product (ubRMSE of ~3

K). The 2-month (Figure 7E) and 3-month (Figure 7I) forecasts show a progressively warmer Indian subcontinent but are cooler in the remainder of the domain compared to the 1-month forecast. Furthermore, the ensemble spread (Figure 7F and 7J) and ubRMSE (Figure 7G, 7H, 7K, and 7L) generally increase with increasing lead times, except for the Pakistan region. Notably, the increase in ensemble spread and error with increasing lead time is greatest in India and less pronounced for the Tibetan Plateau.






The mean climatology of precipitation is much wetter in parts of the domain with higher gradients of elevation, with greater than 15 mm/day in the mountain ranges (e.g., Himalayas) and less than 5 mm/day for other parts of the domain (Figure 8A). Furthermore, for these same regions, the ensemble spread of PRECTOT is also much higher compared to other parts of the domain (Figure 8B), with a mean ensemble spread up to 6 mm/day. The comparisons with MERRA-2 (Figure 8C) and

APHRODITE (Figure 8D) both show a similar magnitude of error throughout most of the domain. The largest errors in PRECTOT forecasts are in the Indian subcontinent and in the Himalayas (ubRMSE up to 5 mm/day). The 2-month (Figure 8E) and 3-month (Figure 8I) forecasts show a drier Indian subcontinent and are somewhat wetter in regions with high elevation when compared to the 1-month forecast. This difference tends to propagate to other variables shown in later figures, i.e., higher fSCA and SWE values in the mountains and lower SM and TWS values in the Indian subcontinent for 2-

and 3-month compared to 1-month forecasts. Furthermore, the ensemble spread is generally higher in the mountain regions and lower over the Indian subcontinent with increasing lead time (Figure 8F and 8J). For the error, there are regions with higher error in forecasts with longer lead times, such as in India (Figure 8G, 8H, 8K, and 8L).

### 3.4.2 Evaluation of Snow Cover Area and Snow Water Equivalent

Snow cover is generally only found in the regions of the domain with high elevation (Figure 9A), and there is much more

snow-covered area in the northwestern parts of the domain (e.g., Hindu Kush and Karakoram). The ensemble spread of fSCA (Figure 9B) is high for much of the domain where there is snow cover, including the Himalayas and the Inner Tibetan Plateau. The 2-month (Figure 9E) and 3-month (Figure 9I) forecasts show higher amounts of fSCA for much of the domain compared to the 1-month forecasts (Figure 9A), which could be attributed to the fact that at longer lead times the forecasts are colder and wetter at higher elevations (Figures 7-8 panels A, E, and I). The comparison with MERRA-2 (Figure 9C) and

MODIS (Figure 9D) both show that errors are present where there is snow cover, where the grid cells that have no snow cover are masked out. The error compared to MERRA-2 (ubRMSE close to 0.2) is noticeably higher than the error compared to MODIS (ubRMSE close to 0.1), especially for regions with high fSCA. This shows that GEOS-S2S fSCA is closer to what is shown in MODIS than to the MERRA-2 product, which supports the results from Figure 4C. Additionally, the ensemble spread (Figure 9F and 9J) and the forecast errors (Figure 9G, 9H, 9K, and 9L) generally increase in the mountain

regions with increasing lead time.

Similarly, the mean climatology of SWE (Figure 10A) indicates that snow is present in the regions of the domain with high elevation, specifically in the major mountain ranges. Consequently, the ensemble spread of SWE (Figure 10B) is also high in these locations (mean spread up to 0.05 m) and very low elsewhere in the domain (mean spread less than 0.01 m). The 2-

month (Figure 10E) and 3-month (Figure 10I) forecasts show higher amounts of SWE in the major mountain ranges, which again could be attributed to the fact that at longer lead times the forecasts are colder (Figure 7E and 7I) and wetter (Figure 8E and 8I) in regions with high elevation gradients. The ubRMSE maps vs. MERRA-2 (Figure 10C) and HMA-SR (Figure 10D) show that errors are higher where there is more snow, which is expected. Here, the error compared to HMA-SR is





considerably higher (ubRMSE up to 0.1 m) than the error compared to MERRA-2 (ubRMSE up to 0.04 m), especially for
regions with high SWE. And like fSCA, the ensemble spread (Figure 10F and 10J) and the forecast errors (Figure 10G, 10H,
10K, and 10L) are generally higher with increasing lead times, particularly in the major mountain ranges.

### 3.4.1 Evaluation of Soil Moisture and Terrestrial Water Storage

The mean climatology of SM (Figure 11A) shows that soil moisture is high in India and Southeast Asia (~0.4 $m^3/m^3$) and is
low in the western and northern parts of the domain (~0.1 $m^3/m^3$). There are lower SM values for forecasts with increasing
lead times for the Indian subcontinent (Figure 11E and 11I). This could be attributed to the fact that at longer lead times the
forecasts are hotter (Figure 7E and 7I) and have less precipitation (Figure 8E and 8I) across the Indian subcontinent.
However, for Myanmar and Southeast Asia, longer lead times produce higher SM values. The ensemble spread of SM
(Figure 11B) is lower for the 1-month forecasts and increases in magnitude for longer lead times (Figure 11F and 11J). The
ubRMSE maps vs. MERRA-2 (Figure 11C) and ESA-CCI (Figure 11D) report higher errors over regions with higher soil
moisture values (ubRMSE of up to 0.06 $m^3/m^3$), and the error increases with lead time (Figure 11G, 11H, 11K, and 11L),
especially in India when compared to MERRA-2 (Figure 11G and 11K). However, when compared to ESA-CCI, the forecast
error decreases with lead time for the western and northern parts of the domain (figure 11H and 11L). Additional data gaps
are shown in these figures due to snow covered and frozen grid cells being masked out in the S2S forecasts and due to
quality control applied during post-processing of the ESA-CCI product.


The mean climatology of TWS (Figure 12A) shows that water storage is higher in Myanmar and Southeast Asia and is lower
in the other parts of the domain. The ensemble spread of TWS (Figure 12B) is higher in the regions with high elevation
gradient (e.g., Himalayan Mountain range). Additionally, the spread of TWS is lower for the 1-month forecasts and increases
in magnitude with lead time (Figure 12F and 12J). The evaluation of GEOS-S2S forecasts of TWS show that the forecasts
are much closer to MERRA-2 (Figure 12C, ubRMSE less than 0.1 m) than to GRACE (Figure 12D, ubRMSE up to 0.3 m).
The errors compared to GRACE are 3-4 times higher in many regions, especially for the Indian subcontinent, Myanmar, and
Southeast Asia (Figure 12D). When compared to MERRA-2, forecasts with longer lead time (Figure 12G and 12K) have
higher errors, yet when compared to GRACE, there is no consistent change in the error with longer lead times (Figure 12H
and 12L), with some regions such as in India having less error with longer lead times.

## 4 Discussion

### 4.1 the Role of Model Initialization and Hydrologic Persistence

S2S forecasting for HMA is in its infancy. Skill has historically been somewhat low and, as shown in our results, certain
variables have high forecast skill while others are more difficult to forecast. When comparing the S2S forecasts with
MERRA-2, Figures 2A and 3A show that the snow variables, SM, and TWS have increased skill at early lead time (1-





month), and for SM and TWS, this skill can persist for forecasts at longer lead time (2-3 months). This could be because GEOS-S2S and MERRA-2 have similar land conditions during initialization, both modeling systems are quite similar, and because these variables have longer persistence and memory in the physical system. When evaluating the S2S forecasts against MERRA-2, forecast skill is highest in long-memory variables (snow and soil moisture related) and lower in near surface atmospheric variables (T2M and precipitation). In all instances, forecast skill decreases rapidly with increasing

forecast lead time. When comparing the S2S forecasts with reference data products (Figures 2B and 3B), the decline in forecast skill across lead times is slower and the anomaly correlations are not consistently statistically different.

Another reason that could explain the skill in certain variables is the role of better land surface initial conditions. For example, fSCA, SWE, SM, and TWS vary more slowly compared to T2M or PRECTOT, and their initial conditions play an

important role in the skill of 1-month forecasts. This can be inferred in our results because, for example, in Figure 2A the forecast skill relative to MERRA-2 is higher for these variables, perhaps due to similar initialization in the GEOS-S2S and MERRA-2 systems. However in Figure 2B, the forecast skill relative to the reference data products is not as high. Enhancements in forecast skill due to improved model initialization for these processes with slower temporal dynamics has been shown in other studies as well (Getirana et al., 2020; Zhou et al., 2021). Therefore, while shorter memory variables

(T2M, PRECTOT) will likely improve the most with improvements in resolution and process representation, gains in forecast skill for longer memory variables will likely be achieved with improved land surface initial conditions, and if successful, increased forecast skill in 1-month lead time will propagate through to longer leads.

## 4.2 The Role of Model Characteristics

### 4.2.1 Resolution

For parts of the domain with high elevation and high topographic variability, many of the variables including PRECTOT, fSCA, SWE, and TWS had large errors (Figures 8C-D, 9C, and 10D) as well as large ensemble spreads (Figures 8B, 9B, 10B, and 12B). This is an indication of the difficulties of accurately forecasting climate for regions of high elevation and complex topography. This could be because of the coarse spatial resolution of the GEOS-S2S simulations with topography posted at a 0.5-degree resolution (i.e., ~ 50 km). The topographic smoothness in the model can impact the simulations in

various ways, such as limited orographic effects or issues associated with the formation and propagation of weather events. To confirm this argument, Cannon et al., (2017) discussed the effects of topographic smoothing on the simulation of winter precipitation in HMA and found that precipitation distributions in topography that is represented in experiments with coarser resolution are biased relative to a simulation with more realistic topography. Furthermore, Zhou et al., (2021) used optimized land initial conditions from GEOS-S2S, and they were able to downscale outputs of soil moisture to 5 km resolution and

assess the forecast time horizon out to 9 months. Therefore, resolution can have a contribution to forecast skill, and it is possible that improved resolution in the S2S forecasts can help to enhance the forecast skill of certain variables.





### 4.2.2 Seasonality: Representing the Monsoon and Other Atmospheric Processes

S2S forecast skill largely depends on getting the seasonal signature in the forecasting system correctly. In our results, there are seasonal patterns in the GEOS-S2S forecast skill (Figures 4-6), and the simulated seasonality and the annual cycle of the
hydrometeorological variables is generally well captured. For example, T2M and PRECTOT errors vary in relation to the Indian monsoon season (JJAS). Precipitation error tends to increase in these months (Figure 5B and 6B) due to higher amounts of precipitation and because monsoon representation in the S2S system is not ideal. T2M error decreases in these months (Figure 5A and 6A) because air temperature is most strongly related to ENSO during the monsoon season (Zhou et al., 2019) and GEOS-S2S tends to capture ENSO rather well (Molod et al., 2020; Hackert et al., 2020; Lim et al., 2021). For
snow variables, fSCA and SWE have low errors when snow is low during the warmest months (Figure 5C and 5D and Figure 6C). An exception is shown in Figure 6D, where forecast errors for SWE are higher during the warm months and lower in the fall, which could be due to the forecasts accumulating SWE more rapidly in the S2S system than what is shown in the HMA-SR reanalysis product. For SM and TWS, error patterns in Figure 5E and 5F and Figure 6E and 6F can primarily be related to monsoon representation in the S2S system, but the errors can also be associated with the observational
difference in the seasonal cycles shown in Figures 4E and 4F. Overall, improving monsoon dynamics in GEOS-S2S will likely improve forecast skill during and following the monsoon season.

Our results confirm those from recent studies, such as Deoras et al. (2021), who compared the predictions of the Indian Monsoon low pressure system in various S2S prediction models on a time scale of 15 days to ERA-Interim and MERRA-2
reanalysis data. Their study found that most models were able to predict basic features, however all S2S models underestimate the frequency of the low-pressure systems and that precipitation biases increased with forecast lead time. Hsu et al. (2021) simulated the East Asian winter monsoon on S2S timescales for 45-day hindcasts using the Model for Prediction Across Scales (MPAS). Their evaluation results revealed that MPAS can simulate the climatological characteristics of the monsoon reasonably, with a surface cold bias of 4% and a positive rainfall bias of 9% over East Asia.
However, they also found that a biased sea surface temperature may modify the circulation over the Western Pacific and affect the simulated occurrence frequency of cold events near Taiwan during winter. Furthermore, climate models are notoriously known to simulate a double Intertropical convergence Zone (ITCZ), in which excessive precipitation is produced on both sides of the equator and especially in the Southern Hemisphere tropics (Hwang and Dargan 2013; Zhang et al., 2019). This is a problem that has been persistent in climate model simulations and can impact the results of S2S forecasts in
the HMA region.



### 4.2.3 Representation of Land Processes

Differences in the level of the S2S forecast skill relative to MERRA-2 and to the other reference products (Table 2 and Figure 2) could be due to certain physical processes that are seen in the signatures of the reference data products but under-represented in the frameworks of GEOS-S2S and MERRA-2. Characterizing hydrometeorological conditions in HMA, through both observations and modeling, is difficult owing to the scarcity of in-situ observations and the complex orographic conditions that impede accurate retrievals of satellite estimates and due to properly representing these processes in the model simulations (Su et al., 2013; Ghatak et al., 2018; Loomis et al., 2019a; Yoon et al., 2019; Gerlitz et al., 2020). These challenges are reflected in the wide range of GEOS-S2S forecast skill when compared to MERRA-2 and reference datasets (i.e., as seen in Figures 2, 4-12).

For example, ESA-CCI data of SM are probably of limited quality in the topographically complex HMA region, and GRACE TWS data shows the signature from rivers, lakes, glacier mass changes, and groundwater pumping that are included in GRACE data but not fully represented in the GEOS-S2S modeling framework. Some regions within HMA, particularly the Indian subcontinent, are known for intense over-pumping of groundwater, which has led to extreme levels of groundwater depletion and has played a prominent role in the loss of freshwater storage for these regions (Tiwari et al., 2009; Xiang et al., 2016; Girotto et al., 2017). This dynamic is captured in the GRACE data but not in the GEOS-S2S forecasts (i.e., compare Figure 7C and Figure 7D). More realistic representation of the various water budget components within GEOS-S2S, such as surface water or groundwater pumping, is likely to contribute to improved skill in the seasonal forecasts.

Appropriate representation of seasonal snow and the antecedent precipitation are critical to realistically forecasting the HMA water cycle. GEOS-S2S forecasts tend to overestimate precipitation relative to both MERRA-2 and the reference observations during all months and nearly all lead times (Figure 4B); this cumulatively impacts snow cover and volume (Figure 4C-D). MERRA-2 corrected precipitation has a known dry bias in precipitation (Figure 4B; Yoon et al., 2019), which limits fSCA and SWE accumulation in the MERRA-2 product (Figure 4C-D). GEOS-S2S is initialized with similar land conditions to MERRA-2, resulting in low fSCA and SWE during winter 1-month lead forecasts; however, GEOS-S2S atmospheric physics increases precipitation as forecasts continue for 2- and 3-month lead forecasts, with more extensive snow cover and higher snow volume (Figure 4C-D), resulting in a seasonal cycle that more closely approximates MODIS and the HMA-SR. This results in a relatively constant regional ubRMSE for all lead times when compared to MODIS and the HMA-SR (Figure 6C-D) and localized improvements in ubRMSE with lead time across the Hindu Kush and Karakoram (Figure 9H-L and Figure 10H-L).




Despite the improvement in the absolute magnitude of snow volume due to increasing precipitation, limitations in the snow
depletion curve used within GEOS-S2S and MERRA-2 result in more extensive snow coverage regionally and more limited
reduction in fSCA relative to SWE in the Hindu Kush (Figure 9H-L and Figure 10H-L). Both GEOS-S2S and MERRA-2
systems use a globally consistent linear relationship between SWE and fSCA with the minimum SWE needed to fully cover
a pixel in snow; that is fSCA=1 if SWE is greater than 26 mm (Stieglitz et al., 2001; Toure et al., 2018). This prescription
was developed based on studies in the northeastern USA and oversimplifies the relationship between SWE and fSCA in
mountainous regions (e.g., Schneider et al., 2021) and results in too much snow cover in the GEOS-S2S forecasts (Figure
4C). Considering the regional pattern of the SWE-fSCA relationship, in addition to improvements in topography (section
4.2.1) and inclusion of regionally important processes like surface albedo evolution, through assimilation (Girotto et al.,
2020) or directly modeling aerosol deposition on snow (Sarangi et al., 2019, 2020), will likely improve snow forecasting and
associated runoff from snow melt within GEOS-S2S

## 4.3 S2S Forecasting for Society's Needs

There are various efforts in the broader community (e.g., Arendt et al., 2017), that are aimed at addressing climate change
impacts on natural hazards (such as flooding or landslides) in the HMA region. Seasonal predictions for HMA from GEOS-
S2S provide useful information for the local populations, for example by potentially providing forecasts with several months
lead time that can be beneficial in preparing for local natural hazards (Bekaert et al., 2020; Stanley et al., 2020). Studies that
utilize numerical methods and state-of-the-art model initialization to enhance S2S prediction skill are beginning to emerge.
For example, Gerlitz et al. (2020) provided a review of seasonal forecasts of water availability in Central Asia. Their review
showed that exceptionally skillful discharge forecasts for the agriculturally relevant vegetation season can be derived by
means of statistical models taking remote sensing-based estimations of snow coverage in the Central Asian mountain regions
as independent covariates, and they found that the consideration of global climate indices, in particular El Niño, allows to
extend the forecast lead-times. Therefore, there is reason to believe that improvements in S2S forecast skill can generally be
achieved.

In our study, the modest levels of forecast error provide a sense of trust in the model forecasts in the context of seasonal
forecasting skill. For example, when compared to MERRA-2, the anomaly correlation for forecasts at 1-month lead was
above 0.18 for all variables and as high as 0.62 (for SM). Relative to the reference data products, the anomaly correlation for
forecasts at 1-month lead was above 0.13 for all variables and as high as 0.24 (for fSCA). Compared to other S2S evaluation
studies, these results for HMA are promising. For instance, de Andrade et al., (2019) showed that anomaly correlation of
global precipitation forecasts at lead time of 1 to 4 weeks was greatly reduced for a variety of S2S models, and by week 4 the
anomaly correlation was consistently below 0.2 for all models. This skill level is comparable to the results presented here for
the GEOS-S2S forecasts in the HMA region. Therefore, the GEOS-S2S forecasts for HMA shown in our study generally
have acceptable skill at 1-month lead time compared to other S2S studies. Additionally, other than looking at the forecast



error to determine whether a forecast was skillful or not, the spread of the forecast ensemble is another metric that gives indication of reliability when preparing for impacts of weather events. For instance, a smaller spread in the S2S forecasts for a given region might be an indication of higher skill for that variable in that region. The results shown in this study provide a benchmark of information regarding the forecast skill as well as the ensemble spread in the GEOS-S2S seasonal forecasts.

Results shown here, and from the GEOS-S2S system in general, can help the community benchmark the S2S forecasting skill for the HMA region, and can also help the community synthesize areas of model improvements that can potentially enhance the forecast skill or expand the time horizon of skillful forecasts. Other areas of enhancing the S2S forecasts could be achieved by the assimilation of land surface observations during the initialization period, for variables such as surface soil moisture or snow-covered area. More accurate representation of initial conditions could lead to improved forecast accuracy at the 1-month lead time, but it is possible that a gain in skill can persist for 2- and 3-month lead times, and perhaps longer. Given the confluence of water resource needs from the local population and the complexity of the hydrologic cycle in HMA, further investment for improving S2S forecasts can be extremely useful for this region, and such improvements can potentially be felt globally.

## 5 Conclusions

We showed here an evaluation of the GEOS-S2S seasonal forecasting system in the HMA region, utilizing various products such as reanalysis data as well as data sets obtained from satellites or model data fusion products. The hydrometeorological variables in our evaluation results included 2-m air temperature (T2M), total precipitation (PRECTOT), fractional snow cover area (fSCA), snow water equivalent (SWE), surface soil moisture (SM), and terrestrial water storage (TWS). The main data product used for the evaluation was the MERRA-2 reanalysis product, which provided information to compare all the considered variables in GEOS-S2S. For further verification, we used separate data for the evaluation of each variable, including ERA5 for T2M, APHRODITE for PRECTOT, MODIS for fSCA, HMA-SR for SWE, ESA-CCI for SM, and GRACE for TWS. We showed various aspects of the model evaluation, such as the skill based on variables, lead time, or observation used for the evaluation. We also displayed the climatology of the GEOS-S2S ensemble mean, the ensemble spread, and the mean error for each variable.

Choice of evaluation datasets heavily impacted our results. For example, when compared to MERRA-2, variables with longer memory in the physical climate system, such as soil moisture and TWS, had higher accuracy in the seasonal forecasting system compared to variables representing quickly changing processes, such as temperature or precipitation. This was true when comparing the seasonal forecasts to the MERRA-2 reanalysis because of similar initialization and model architecture as used in GEOS-S2S. However, this finding was not conclusive when reference data products were used in the evaluation. Finally, we provided potential avenues for model improvements that can help enhance the forecasts, such as higher resolution topography representation as well as more realistic representation of surface water and groundwater

pumping. Other paths to improvement could be the assimilation of observations for the initialization of land surface state variables, such as soil moisture or snow cover.

Our results shown here benchmark the GEOS-S2S system's ability to forecast HMA on the seasonal timescale. We showed that, when compared to MERRA-2, the anomaly correlation for forecasts at 1-month lead was above 0.18 for all variables

and as high as 0.62 (for SM). Relative to the reference data products, the anomaly correlation for forecasts at 1-month lead was above 0.13 for all variables and as high as 0.24 (for fSCA). Compared to other S2S evaluation studies, these results for HMA are promising. The reported results should motivate future improvements in the forecasts, such as model initialization, model physics, or more realistic orographic representation, that will be helpful for climate adaptation, natural hazard mitigation, and water resources planning for the population of HMA.

**6 Code Availability**

MATLAB code and other scripts to process the GEOS-S2S outputs are available on request, and scripts to plot and evaluate the data and produce all analysis in this study are available upon request from Dr. Massoud.

**7 Data Availability**

The GEOS-S2S-V2 data is available on the Discover server of NCCS. GEOS-S2S-V2 forecast output data are presently

available at https://gmao.gsfc.nasa.gov/gmaoftp/gmaofcst/. The file specification document that elaborates on the available output from GEOS-S2S is available online (from https://gmao.gsfc.nasa.gov/pubs/docs/Nakada1033.pdf). MERRA-2 data can be downloaded at no cost from (https://disc.gsfc.nasa.gov/datasets?project=MERRA-2), ERA5 data from (https://cds.climate.copernicus.eu/cdsapp#!/dataset/reanalysis-era5-single-levels-monthly-means?tab=overview),  MODIS data from (https://modis.gsfc.nasa.gov/data/), HMA-SR data from (https://nsidc.org/data/HMA_SR_D/versions/1), ESA-CCI

from (https://esa-soilmoisture-cci.org/), and GRACE data from (https://earth.gsfc.nasa.gov/geo/data/grace-mascons).

**8 Author Contribution**

Dr. Massoud, Dr. Andrews, Dr. Girotto – Conceptualization, Data Curation, Formal Analysis, Investigation, Methodology, Writing – original draft preparation, Writing – review and editing, Visualization, Validation. Dr. Kim and S. Ruehr – Data Curation, Writing – review and editing. Dr. Reichle and Dr. Molod – Supervision, Writing – review and editing.

**9 competing Interest**

The authors declare that they have no conflict of interest.



## 10 Acknowledgements

The authors acknowledge the NASA HiMAT team for funding this work (GRANT # 80NSSC20K1301), as well as for generous data sharing and broader discussions that helped shape the paper. GMAO's GEOS-S2S-V2 development was
funded under the NASA Modeling, Analysis, and Prediction program GMAO "core" funding. Computational resources were provided by the NASA High-End Computing (HEC) Program through the NASA Center for Climate Simulation (NCCS) at the Goddard Space Flight Center and the NASA Advanced Supercomputing (NAS) division. The authors thank Agniv Sengupta for information on and preparation of the topography map.

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



**Figures**



**Figure 1: Topography and ocean bathymetry using the NOAA National Geophysical Data Center's ETOPO1 Global Relief Model. The map shows the elevation (m) for the HMA domain. The topography shown in this map is not the same as the topography used by GEOS-S2S, which has a coarser representation of the actual topography in the HMA region. In this figure, countries are shown in black text, mountain ranges in white, and main rivers that are in major basins in blue.**






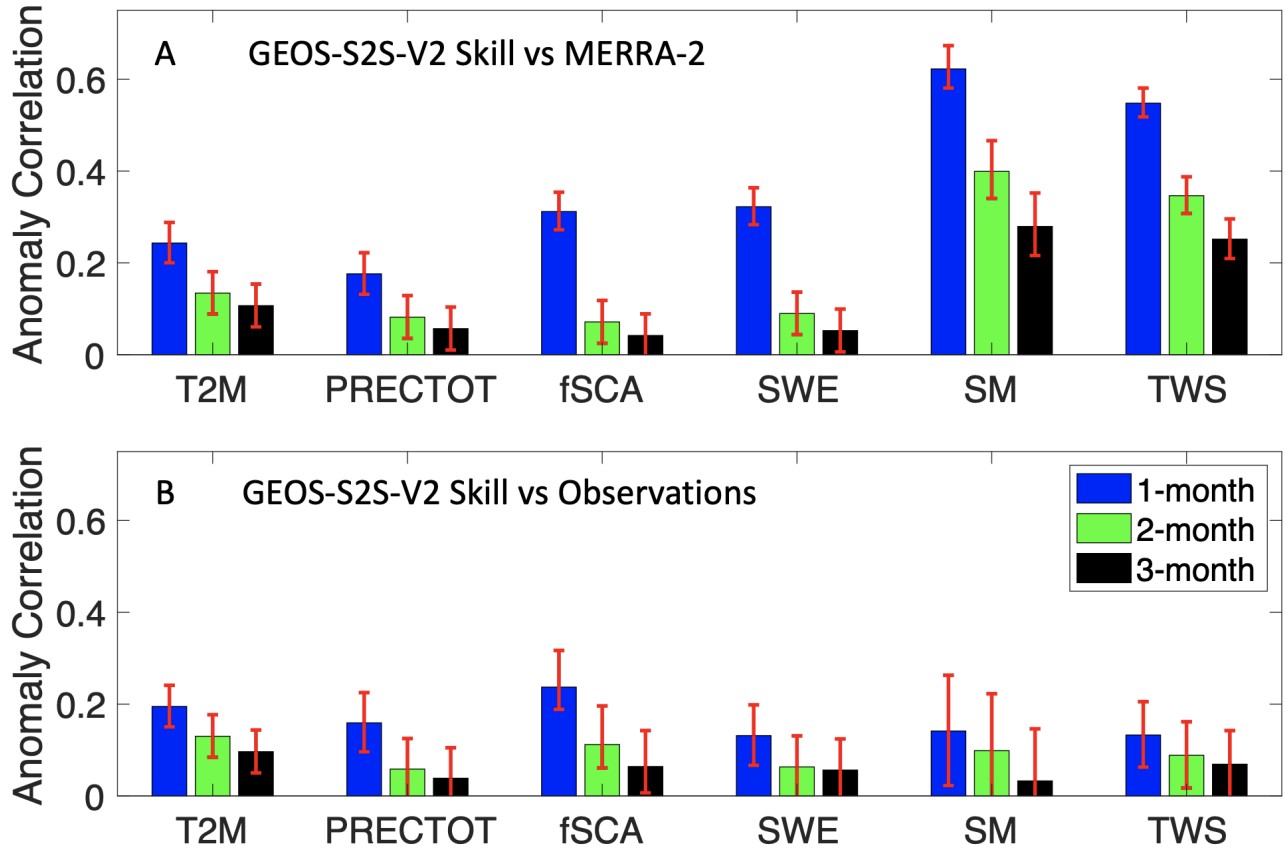

**Figure 2: Anomaly correlation skill between variables for the GEOS-S2S forecasts when evaluated against MERRA-2 (Figure 2A) and against reference data products (Figure 2B). The evaluation of the 1-month lead forecasts is shown in the first bar (blue), 2-month in the second bar (green), and 3-month in the third bar (black). The red error bars indicate the spread of the spatially averaged anomaly correlation for each variable. The reference data that are used in Figure 2B are listed in Table 1.**





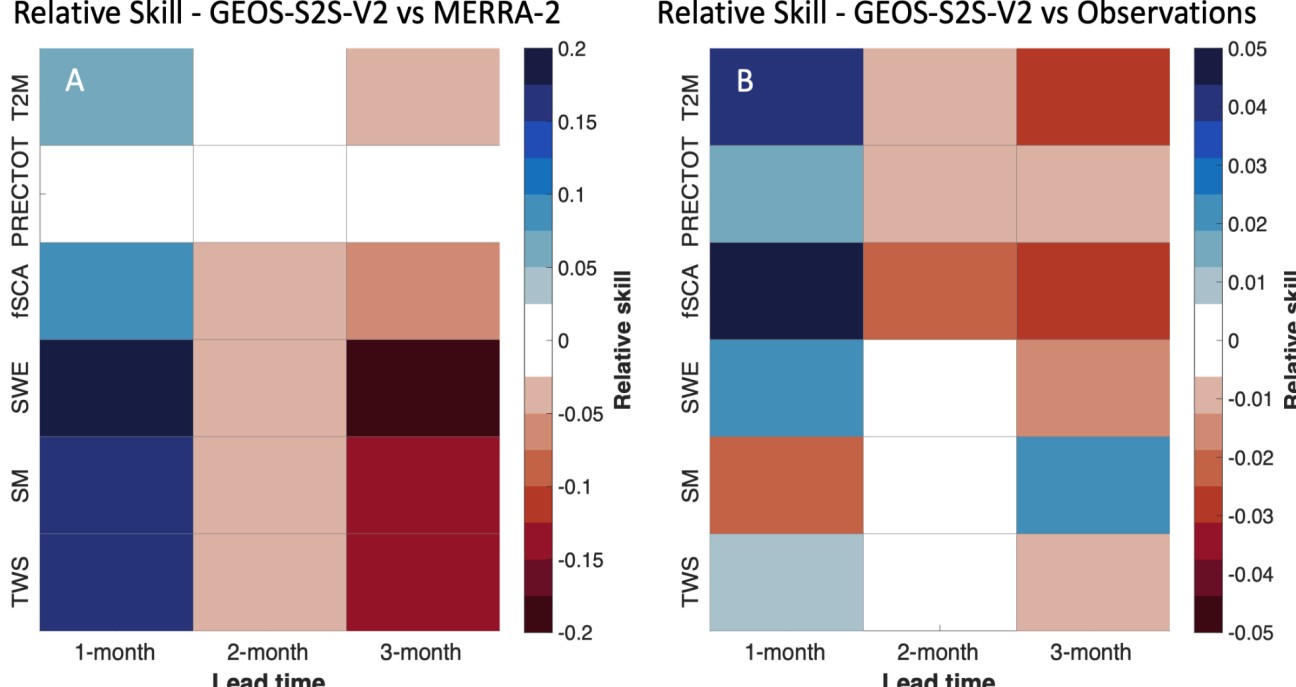

Figure 3: Portrait diagrams visually depicting the difference in skill between variables and between forecast lead time. Figure 3A shows the Relative Skill when comparing the GEOS-S2S forecasts with the MERRA-2 reanalysis, and Figure 3B shows this when comparing the forecasts with reference data products. Here, 'Relative Skill' refers to the ubRMSE of each variable at a given lead time, normalized by the average ubRMSE across all lead times. The equation to estimate the 'Relative Skill', or $E^*$, is ( [ $E^* = (E - E_m) / E_m$ ] ), where $E$ represents the skill at a specific lead time and $E_m$ is the mean skill from all lead times. Blue depicts higher and red indicates lower 'Relative Skill'. Note the difference in the bounds of the colorbar between Figures 3A and B.



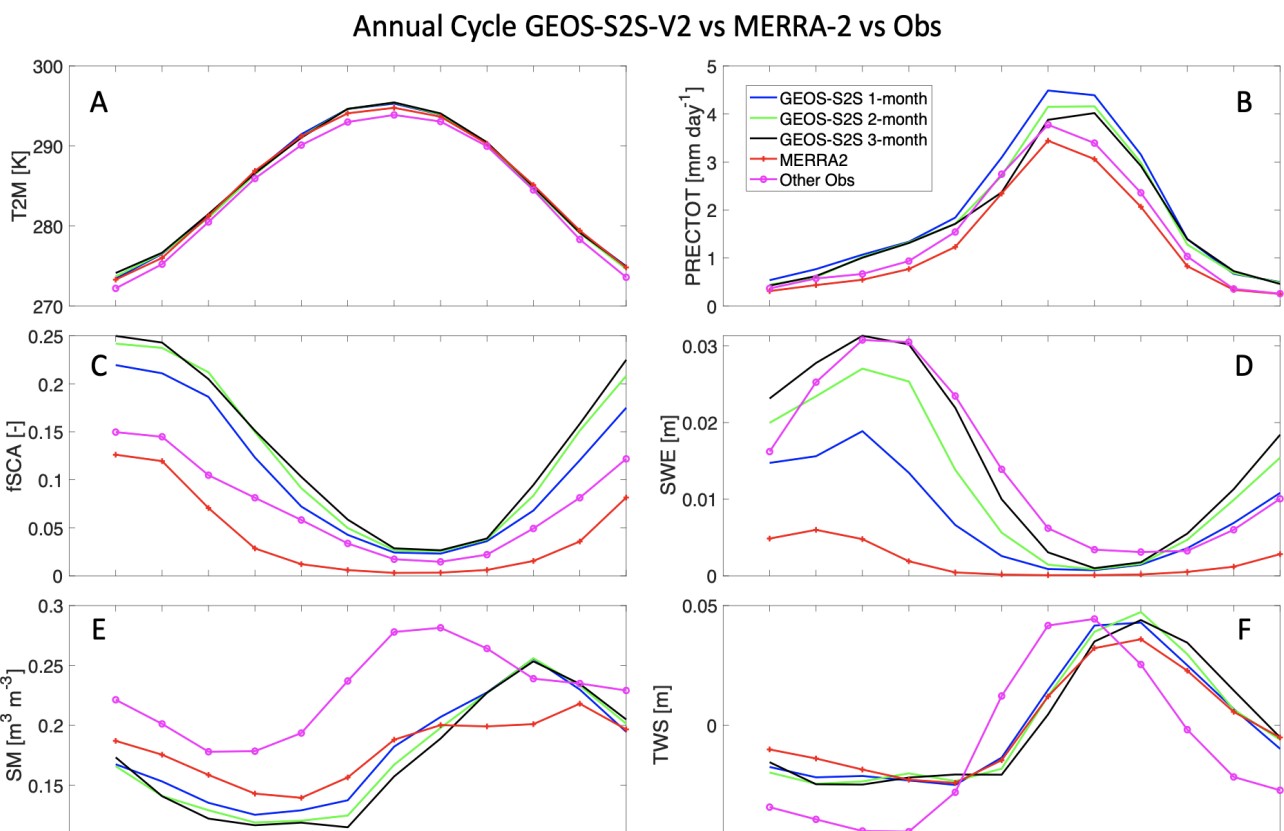

**Figure 4: Annual cycle for each variable, averaged over the HMA domain. The annual cycles from the GEOS-S2S forecasts are shown for all lead times (blue, green, and black curves), and those estimated from the MERRA-2 reanalysis (red) and the reference data products (pink) are shown for comparison.**






**Figure 5: The expected error (ubRMSE) based on which month is forecasted. Shown here are results for 1-month (blue, first bar), 2-month (green, second bar), and 3-month (black, third bar) lead times for each variable. For example, the three bars for the month of April include the 1-month, 2-month, and 3-month forecasts, which are the forecasts initialized in March, February, and January respectively. The results displayed in this figure use MERRA-2 as the evaluation target.**



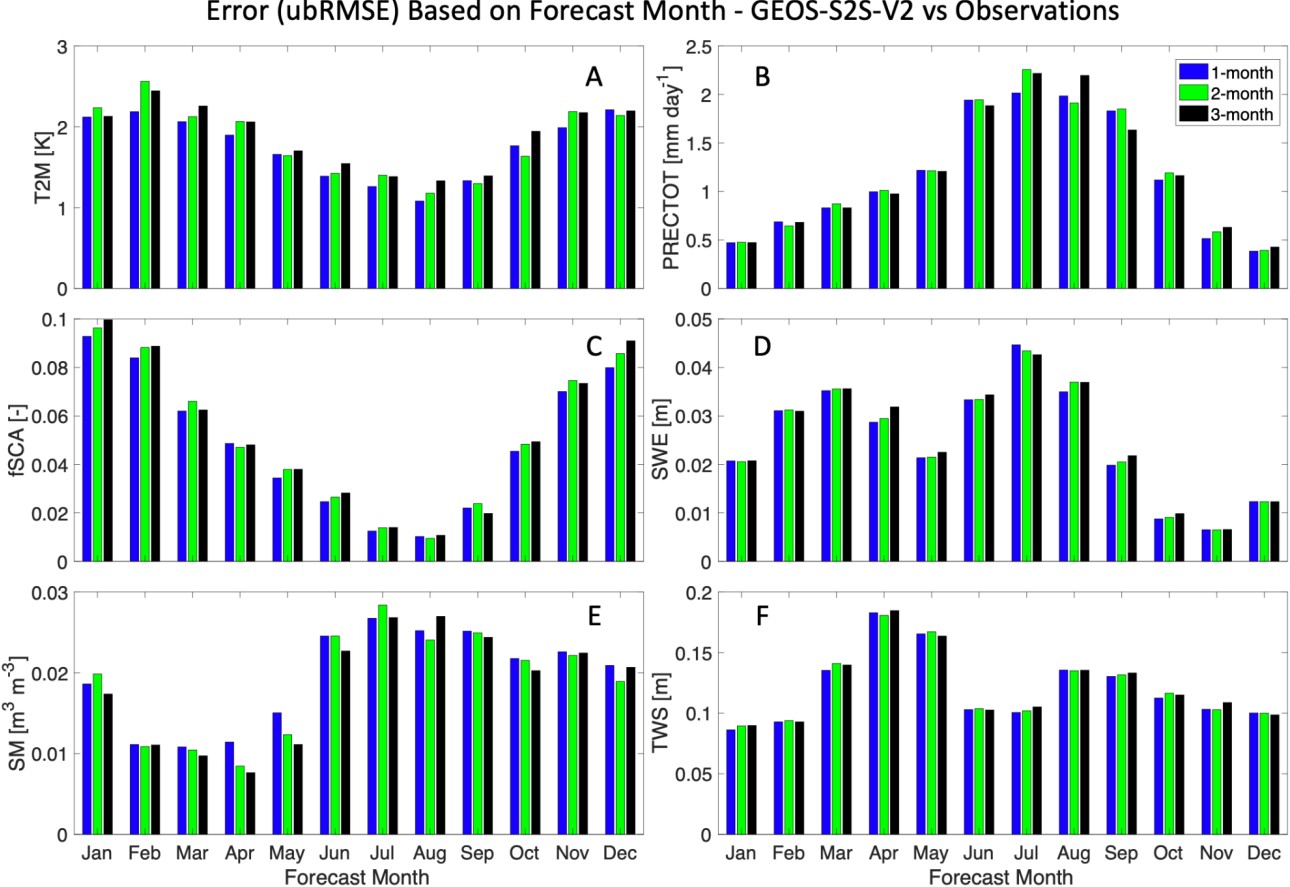

**Figure 6: Same as Figure 5, but the results displayed in this figure use the reference data products as the evaluation target. The data that are used in this figure are: ERA5 for T2M, APHRODITE for PRECTOT, MODIS for fSCA, HMA-SR for SWE, ESA-CCI for SM, and GRACE for TWS.**




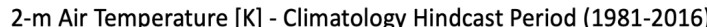

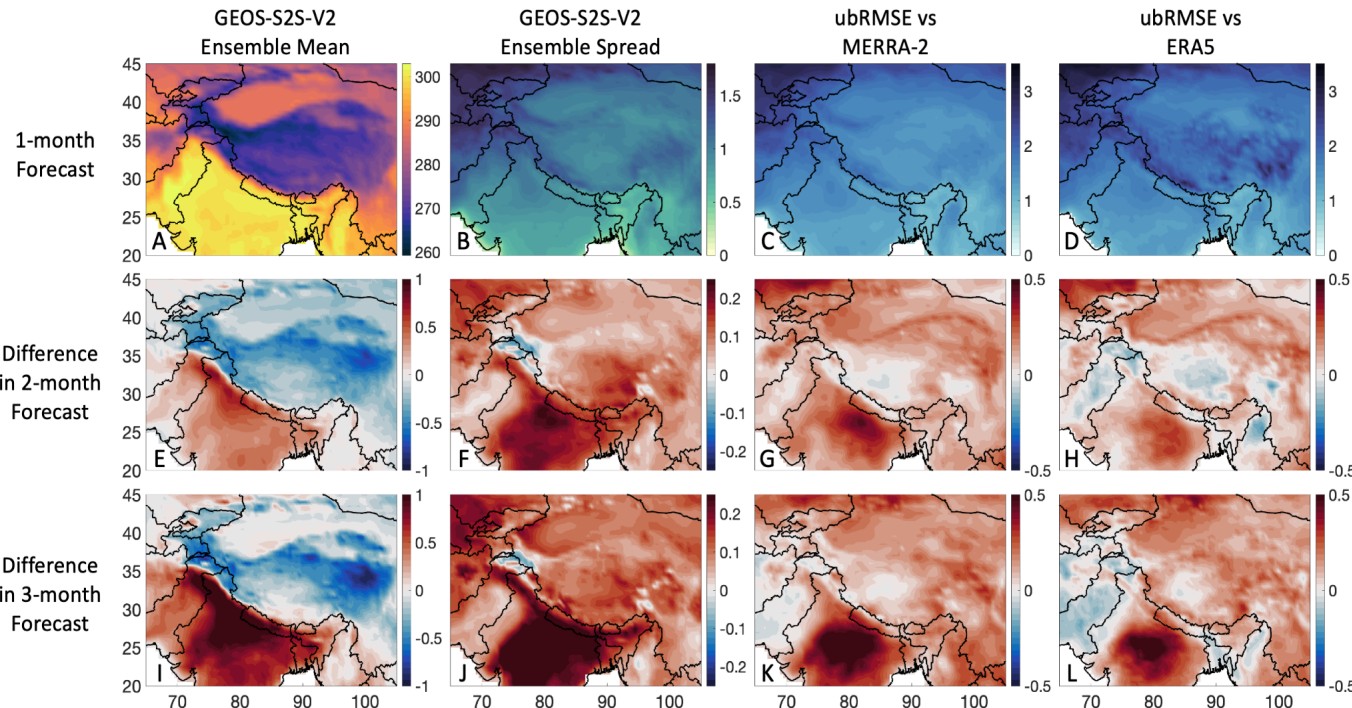

**Figure 7: Screen-level (2-m) air temperature (T2M) metrics in HMA. Figure 7A shows the climatology (long term mean) from the**
**GEOS-S2S 1-month forecast for the hindcast period (1981-2016). Figure 7B shows the ensemble spread from the GEOS-S2S 1-month forecast, calculated as the standard deviation of the model ensemble at each grid cell. Figures 7CD show the ubRMSE when comparing the GEOS-S2S 1-month forecast to MERRA-2 and ERA5, respectively. The bottom two rows of figures show the differences in the climatology, ensemble spread, and ubRMSE between the 2-month (Figures 7E-H) and 3-month (Figures 7I-L) forecasts compared to the 1-month forecast shown in the top row. Note, to calculate the difference shown in the bottom two rows,**
**the 1-month maps in the top row are subtracted from the corresponding 2- and 3-month maps (i.e., 2-month maps minus 1-month maps and 3-month maps minus 1-month maps, respectively). Therefore, red in the subfigures indicates higher values (i.e., hotter temperature, larger spread, or larger error) in the 2- and 3-month forecasts, and blue indicates lower values compared to the 1-month forecasts. The units for these plots are in [K].**



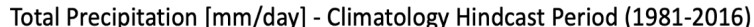

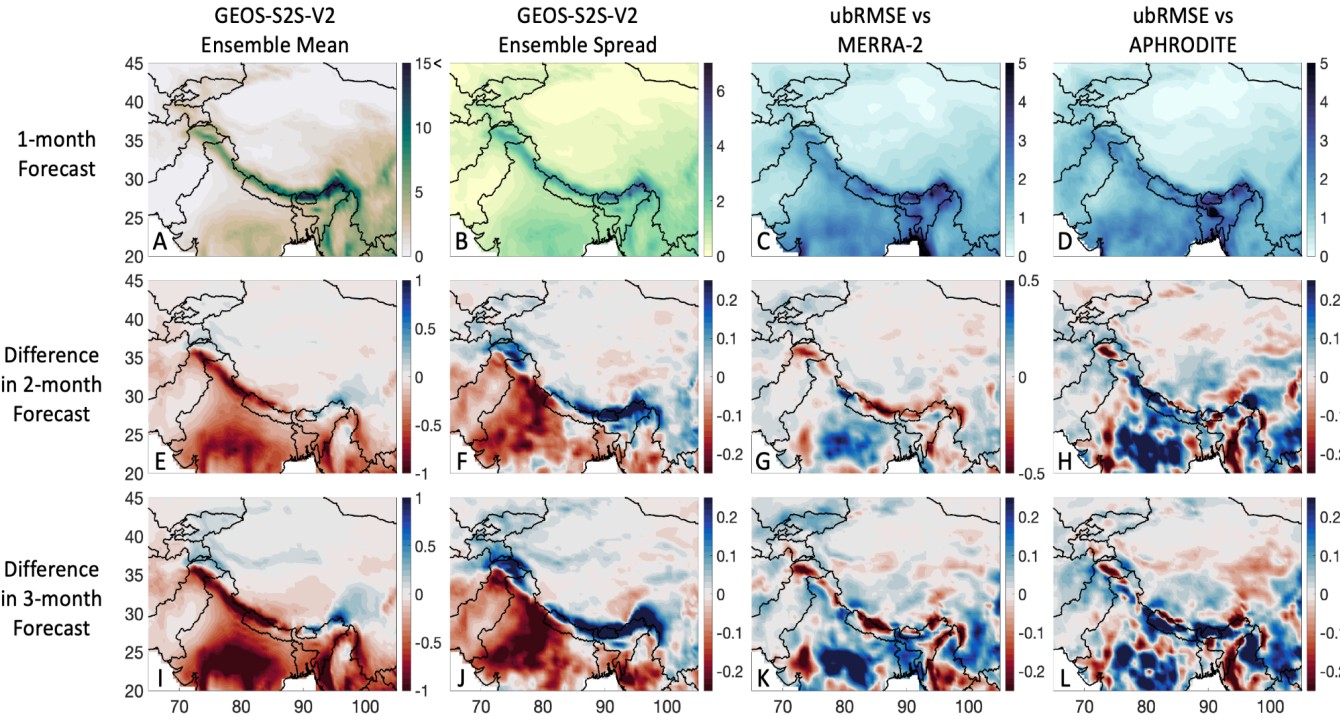

**Figure 8: As in Figure 7, but for precipitation (PRECTOT) in [mm/day] and vs. APHRODITE in the right column. Here, red in the subfigures indicates lower values (i.e., less precipitation, smaller spread, or smaller error) in the 2- and 3-month forecasts and blue indicates higher values compared to the 1-month forecasts.**






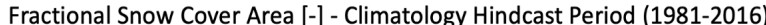

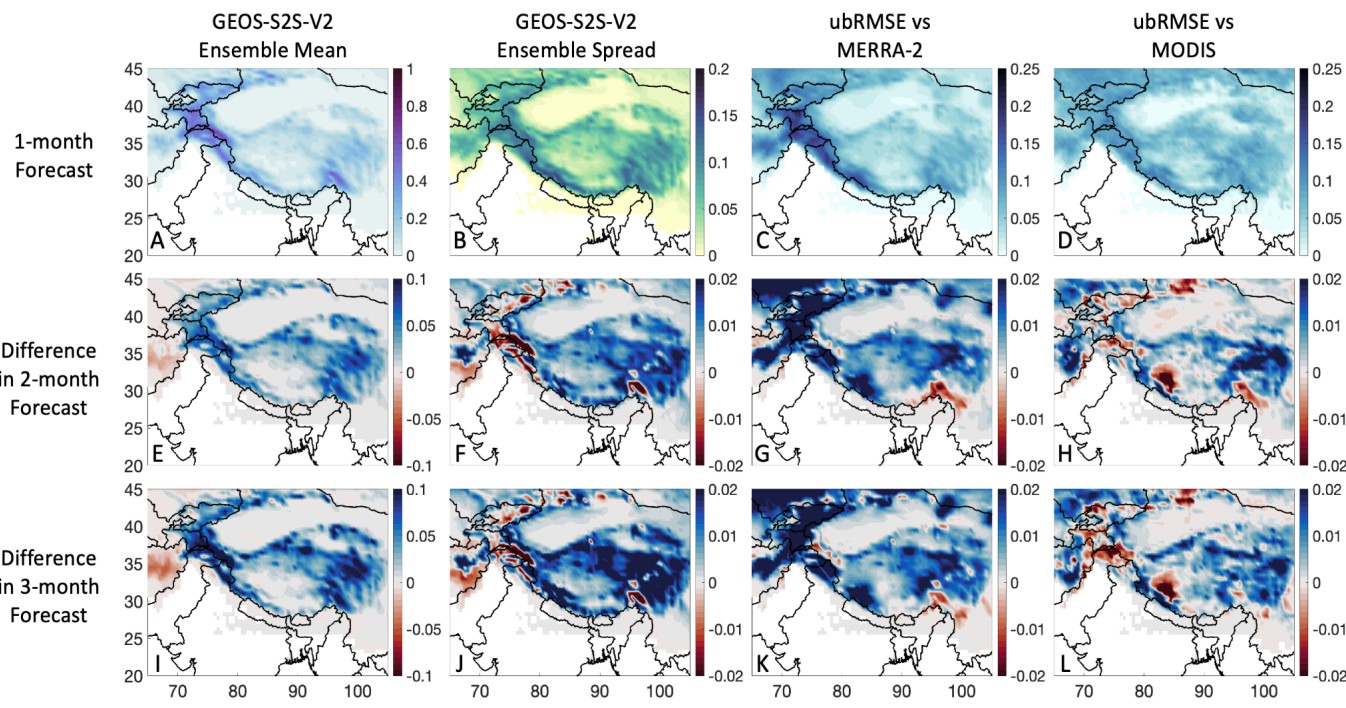

145

**Figure 9:** As in Figure 7, but for fractional snow cover area (fSCA) [unitless] and vs. MODIS in the right column. Grid cells that are masked out (in white) show areas with no-data-values. Here, red in the subfigures indicates lower values (i.e., less snow cover, smaller spread, or smaller error) in the 2- and 3-month forecasts and blue indicates higher values compared to the 1-month forecasts.

150



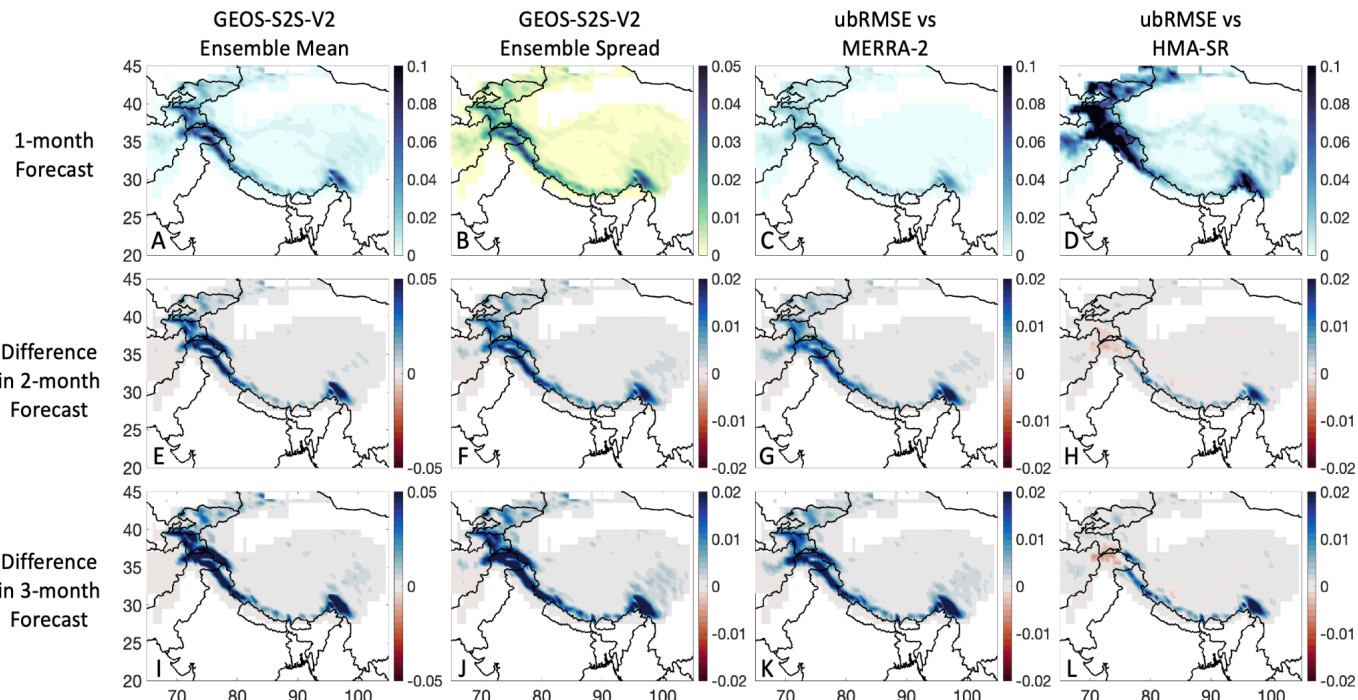

**Figure 10:** As in Figure 7, but for snow water equivalent (SWE) in [m] and vs. HMA-SR in the right column. Grid cells that are masked out (in white) show areas with no-data-values. Here, red in the subfigures indicates lower values (i.e., less snow water, smaller spread, or smaller error) in the 2- and 3-month forecasts and blue indicates higher values compared to the 1-month forecasts.

1155



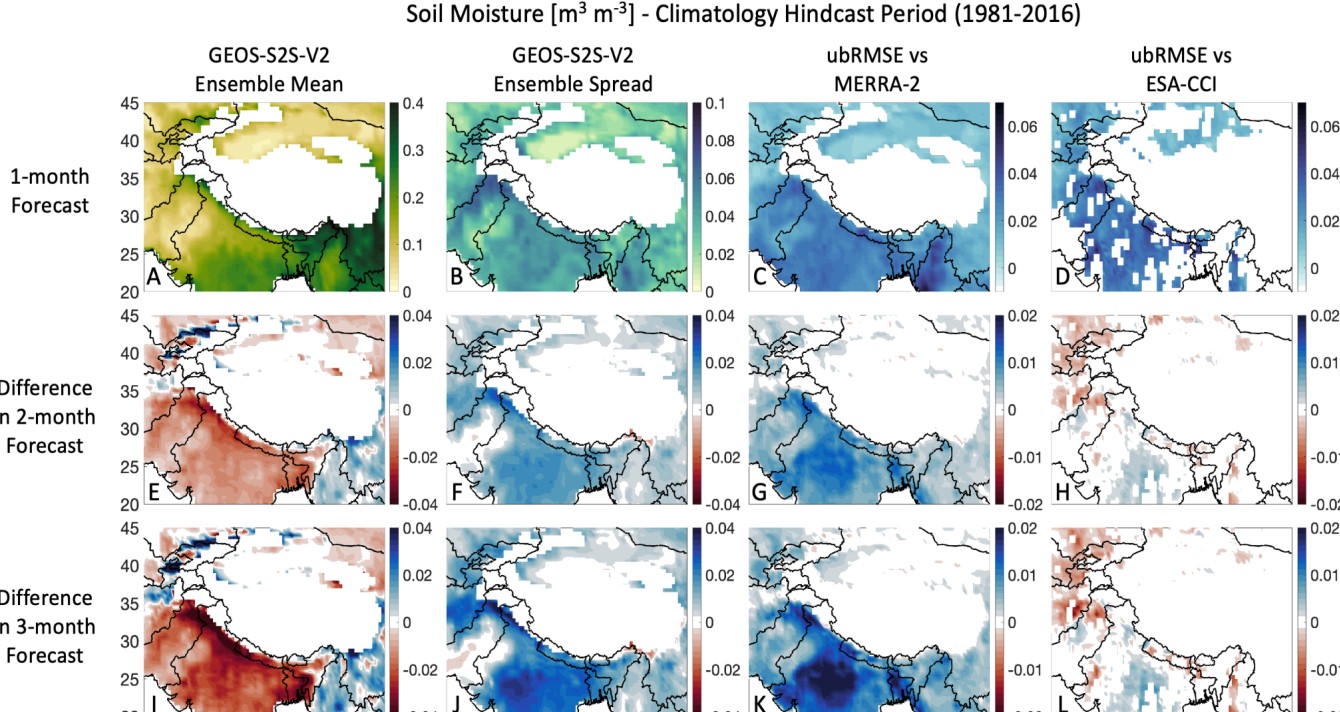

**Figure 11: As in Figure 7, but for soil moisture (SM) in [m3/m3] and vs. ESA-CCI in the right column. Grid cells that are masked out (in white) show areas with no-data-values. Here, red in the subfigures indicates lower values (i.e., less soil moisture, smaller spread, or smaller error) in the 2- and 3-month forecasts and blue indicates higher values compared to the 1-month forecasts.**

1160



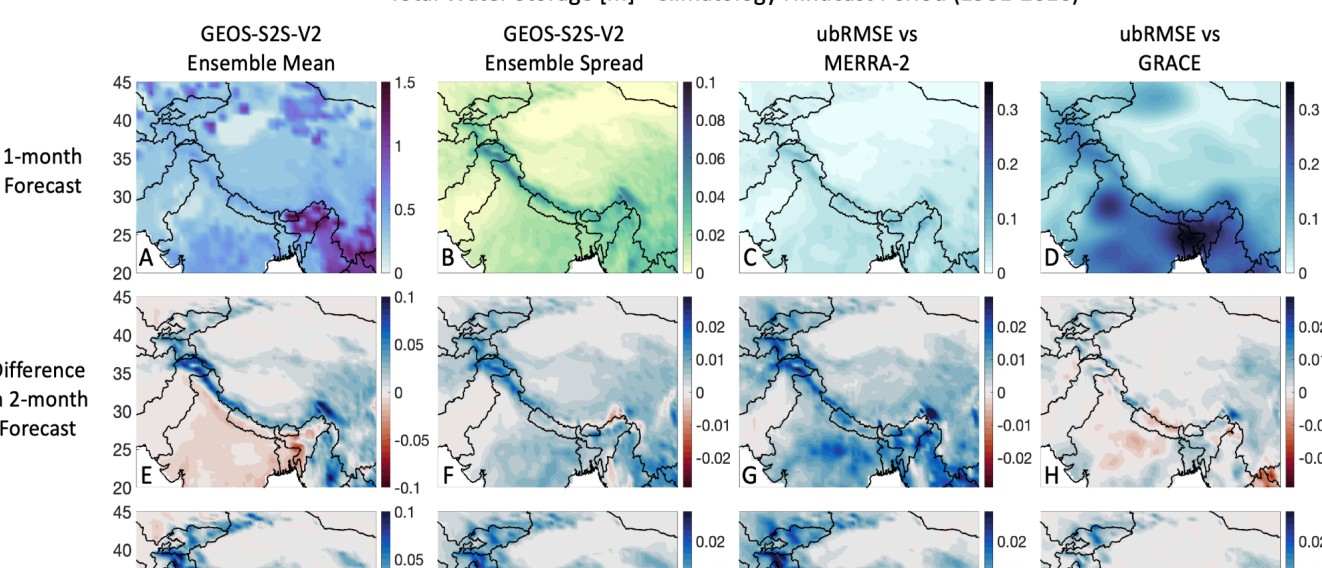

**Figure 12: As in Figure 7, but for terrestrial water storage (TWS) in [m] and vs. GRACE in the right column. Here, red in the subfigures indicates lower values (i.e., less TWS, smaller spread, or smaller error) in the 2- and 3-month forecasts and blue indicates higher values compared to the 1-month forecasts.**



**Tables**

**Table 1: The list of all data products used, including the seasonal forecasting system GEOS-S2S-V2, the MERRA-2 reanalysis product, and the various reference data products. The information in this table includes the period of data availability, the period used in the evaluation, the variables used in this study, the original spatial and temporal resolution, and the main reference for each data set. GEOS-S2S-V2, MERRA-2, and ERA5 data are provided up until the present day and production of these data sets occurs in near-real-time, where quality-assured monthly updates are typically published within 3 months of data production. GRACE data is originally provided at 3 degrees spatial resolution, but the version used here is posted at 1-degree spatial resolution.**

| Data Product | Available period | Evaluation period | Variables Evaluated | Spatial Resolution | Temporal Resolution | Reference |
|---|---|---|---|---|---|---|
| GEOS-S2S-V2 | 01/1981-12/2021 | 01/1981-12/2016 | All variables | 0.5 Degrees | Daily | Nakada et al., 2018 |
| MERRA-2 | 01/1980-12/2021 | 01/1981-12/2016 | All variables | 0.625x0.5 Degrees | Hourly | Gelaro et al., 2017 |
| ERA5 | 01/1979-12/2021 | 01/1981-12/2016 | T2M | 31 kilometers | 3 hours | Hersbach et al., 2020 |
| APHRODITE | 01/1998-12/2015 | 01/1998-12/2015 | PRECTOT | 0.05 Degrees | Daily | Yatagai et al., 2012 |
| MODIS | 02/2000-12/2016 | 02/2000-12/2016 | fSCA | 0.05 Degrees | Daily | Hall et al., 2002 |
| HMA-SR | 10/1999-09/2017 | 01/2000-12/2016 | SWE | 500 meters | Daily | Liu et al., 2021b |
| ESA-CCI | 01/1978-12/2020 | 01/2000-12/2016 | SM | 0.25 Degrees | Daily | Dorigo et al., 2017 |
| GRACE | 04/2002-10/2017 | 04/2002-12/2016 | TWS | 3 Degrees | Monthly | Tapley et al., 2004 |



Table 2: The unbiased RMSE (ubRMSE) and the anomaly correlation (Ranom) for all variables when comparing the GEOS-S2S forecasts to the reanalysis ('MERRA-2') and the reference data products ('Reference data'). The reference data that are used here are: ERA5 for T2M, APHRODITE for PRECTOT, MODIS for fSCA, HMA-SR for SWE, ESA-CCI for SM, and GRACE for TWS (Section 2.3).

| | | RMSE | | | Ranom | |
| --- | --- | --- | --- | --- | --- | --- |
| **GEOS-S2S vs MERRA2** | **1-month** | **2-month** | **3-month** | **1-month** | **2-month** | **3-month** |
| **T2M [K]** | 1.61 | 1.74 | 1.77 | 0.24 | 0.13 | 0.11 |
| **PRECTOT [mm/day]** | 1.06 | 1.08 | 1.08 | 0.18 | 0.08 | 0.06 |
| **fSCA [-]** | 0.035 | 0.041 | 0.041 | 0.31 | 0.07 | 0.04 |
| **SWE [m]** | 0.002 | 0.003 | 0.004 | 0.32 | 0.09 | 0.05 |
| **SM [$m^3/m^3$]** | 0.019 | 0.023 | 0.025 | 0.62 | 0.40 | 0.28 |
| **TWS [m]** | 0.025 | 0.032 | 0.035 | 0.55 | 0.35 | 0.25 |
| | | RMSE | | | Ranom | |
| **GEOS-S2S vs Reference data** | **1-month** | **2-month** | **3-month** | **1-month** | **2-month** | **3-month** |
| **T2M [K]** | 1.78 | 1.87 | 1.90 | 0.19 | 0.13 | 0.10 |
| **PRECTOT [mm/day]** | 1.03 | 1.06 | 1.06 | 0.16 | 0.06 | 0.04 |
| **fSCA [-]** | 0.048 | 0.051 | 0.052 | 0.24 | 0.11 | 0.06 |
| **SWE [m]** | 0.021 | 0.021 | 0.022 | 0.13 | 0.06 | 0.06 |
| **SM [$m^3/m^3$]** | 0.021 | 0.020 | 0.020 | 0.14 | 0.10 | 0.03 |
| **TWS [m]** | 0.101 | 0.103 | 0.104 | 0.13 | 0.09 | 0.07 |