# Peer review of "Seasonal forecasting skill for the High Mountain Asia region in the Goddard Earth Observing System"

_EGUsphere, 2022_

## Author Comment (AC3)

**Final Authors Comments to Reviewer 3**

Comment on egusphere-2022-449
Anonymous Referee #3

Referee comment on "Seasonal forecasting skill for the High Mountain Asia region in the Goddard Earth Observing System" by Elias Charbel Massoud et al., EGUsphere, https://doi.org/10.5194/egusphere-2022-449-RC3, 2022

This paper examines the seasonal prediction skill of the NASA's Goddard Earth Observing System (GEOS) S2S prediction system over High Mountain Asia for a series of hydrological variables. A set of observational data and the MERRA-2 and ERA-5 re-analyses are used as benchmarks.

Monthly means of reforecasts over the period 1981 to 2016 are analysed.

The paper provides a clear and well written description of the system and presents a clear analysis the forecasts of these hydrological variables. It provides a solid detailed reference to further assess more specific aspects of the performance of the GEOS S2S prediction system over HMA. The Figures are clear but, while many detailed descriptions are provided, I find the paper a little thin on new science and on understanding potential sources of predictability. At the very least, some more discussion points should be included.

*Author Comments:* We very much thank the reviewer for the time spent on our manuscript. We will aim to include more discussion points that address this concern in the next version of the paper, including the addition of evaluation for specific sub-regions as well as the addition of new text in various parts of the manuscript.

MAIN COMMENTS

1) The skills over this large, heterogenous region are small, especially when measured against independent data. I wonder if such small skills are relevant at all. The prediction system may have higher skill in some variables, in more limited domains or at certain times of year.

Low skill is not unexpected given the large area, the varied topography (from lowlands to high mountains) and land cover, and the different regional climates of this large region. For example, only a small Southeastern part of the Tibetan Plateau (and hence, only parts of the HMA) is influenced by the Indian summer monsoon (ISM).

It is well-known that the skills depend on the verification data, and they would be higher, in most cases, when verified MERRA-2 given the parent model. It would be of interest to verify some variables against several datasets, or better, against merged datasets that

take into account uncertainties in the various observations. I realise that such merged dataset might not exist over this region, but the point could be mentioned in the Discussion.

*Author Comments:* We thank the reviewer for this insightful comment. We do understand how the skills reported here can seem minor and not relevant, however we would like to point out that the skills shown are based on anomaly skill (e.g., anomaly correlation and unbiased RMSE). Therefore, any skill that is above 0 anomaly correlation is relevant, since it essentially means that it is more skillful than just guessing the mean climate. For S2S predictions at a range of 1-3 months, we believe that this is rather useful and promising.

As for gaining results in the more limited domains, our results in Figures 5-12 show maps of the skill and the ensemble spreads for the whole domain, which can be used to analyze spatial differences in the results and to pinpoint specific regions with higher or lower skills in general. Figures such as these can be used to interpret the skill level for different regions within the domain, such as low- or high- elevation regions. For example, Figure 7 shows that there is a higher ensemble spread as well as a higher error for temperature in the India subregion compared to other regions in the domain. To help further satisfy this concern, we will provide new plots showing the evaluation metrics for smaller sub-regions within the domain and will provide more discussion about these results.

Regarding the use of multiple data sets or perhaps a merged data product, this is the reason we use multiple sources of observed and verification data in our study. Relying on solely one source of information may be misleading, hence we used 2 different products for each climate variable to get a sense of the uncertainty in the forecast skill for each variable. We are not aware of a merged data product for this region, perhaps the MERRA-2 product is a close example of what the reviewer is describing here. In the discussion, we will add statements about the potential of having a data set like this and how it can be helpful for the evaluation.

2)Concerning actual societal needs, the reliability of such forecasts is a question of utmost importance that needs to be addressed in a probabilistic context. Is it possible to quantify the reliability of the forecasts with the current system using standard metrics? At least, the outlook could be mentioned in the Discussion.

*Author Comments:* We thank the reviewer for this comment, and we agree that the reliability or the uncertainty in the reported skill is important especially for societal needs. For this statement, we point to the ensemble spreads shown in Figures 5-12, which show the spread and therefore the 'reliability' of the forecasts from the model. The higher the ensemble spread in these plots, the less certain the various ensemble members are for each climate variable and lead time. Furthermore, the use of multiple data sources in the evaluation also allows us a look at the uncertainty in our results. We will include a statement in the discussion to elaborate on the interpretation of uncertainty and how it impacts societal needs. We will also provide additional text in the discussion section describing the reliability of the forecasts and how that can be estimated using standard reliability metrics. For example, we will provide new figures

showing the comparison between the ensemble spread and the error. Generally, one can compute the spread/error ratio with the goal of that being close to 1; if it is larger than 1 (more spread than error) this is considered "underconfident", and if it is less than 1 this is considered "overconfident" (Fortin et al., 2014). We will provide new plots and relevant references to provide additional and meaningful verification metrics.

3)I wonder about the relationship between surface temperature and the snowpack. Is there is a strong coupling between the two in the forecasts during some months? This could provide a source of skill.

*Author Comments:* We thank the reviewer for this insightful comment. We do not find any direct relationship between the skill in temperature and how it affects the skill in the snowpack. This is an idea that deserves some discussion, and we will provide some additional details in the paper to discuss this.

4) Improved prediction of the circulation could lead to improved skill. The authors mention the importance of the ISM. I believe that wintertime precipitation over the northern part of HMA is brought by the so-called westerly disturbances. The authors could mention in the Discussion, whether the dynamics and the associated with precipitation is well represented in the forecast.

*Author Comments:* We again thank the reviewer for this insightful comment. We do report on the importance of the ISM in Section 4.2.2. We will include some details about westerly disturbances and how properly representing that process within the model can impact the forecast skill.

5)There has been a significant effort in recent years to assess the impact of land initialisation (esp. snow, soil moisture) in S2S and seasonal forecasts and some studies are relevant for the HMA region yet there is little mention of that relevant literature.

Koster, R. D., Mahanama, S. P. P., Yamada, T. J., Balsamo, G., Berg, A. A., Boisserie, M., et al. (2011). GLACE2: The second phase of the global land atmosphere coupling experiment: Soil moisture contribution to subseasonal forecast skill. Journal of Hydrometeorology,12(5), 805–822.

Senan, R., Orsolini, Y.J., Weisheimer, A. et al. Impact of springtime Himalayan–Tibetan Plateau snowpack on the onset of the Indian summer monsoon in coupled seasonal forecasts. Clim Dyn 47, 2709–2725 (2016). https://doi.org/10.1007/s00382-016-2993-y

*Author Comments:* We thank the reviewer for providing these works in the literature. We do report on the impact of land initialization in Section 4.1, and we will also read the works listed here by the reviewer and we will aim to include more comments in the discussion based on these papers.

MINOR COMMENTS

- The words Seasonal forecasts and S2S forecasts seem to be used loosely throughout the paper. The seasonal forecasts are 9-month long but only the first 3 months are analysed. Some operational centers have different set-ups for Seasonal and S2S prediction systems. The authors could double check that S2S is used as it is meant.

*Author Comments:* We thank the reviewer for this comment. From our understanding, subseasonal forecasting refers to predictions at 10 days to 1 month, and seasonal predictions refer to predictions at 1 month to 9 months lead time. We will do a more thorough check about this definition and make sure our paper is consistent in the terminology throughout.

- It was not clear to me whether total precipitation is liquid precipitation or if it contains also solid precipitation.

*Author Comments:* The form of precipitation analyzed in this paper is the PRECTOTCORR variable from the GEOS-S2S system, and is derived from the PRECTOT variable, which is total precipitation including rain and snow, i.e., PRECTOT = liquid + solid (total) precipitation. We will include a comment in the paper to better explain this.

- The information on ensemble size should be presented more clearly (Abstract, or Table)

*Author Comments:* We thank the reviewer for this comment. We will provide a better and more clear description of the ensemble size.

Wording

L58: the foothills of the Himalayas perhaps better than the foot of the Himalayas (?)

*Author Comments:* We thank the reviewer for this comment. We will fix this in the paper.

L560: precipitation is used twice in same sentence.

*Author Comments:* We thank the reviewer for this comment. We will fix this in the paper.

---

## Author Response (AR1)

**Final Authors Comments to Reviewer 1**

Comment on egusphere-2022-449
Anonymous Referee #1

Referee comment on "Seasonal forecasting skill for the High Mountain Asia region in the Goddard Earth Observing System" by Elias Charbel Massoud et al., EGUsphere, https://doi.org/10.5194/egusphere-2022-449-RC1, 2022

Review of Massoud et al. entitled "Seasonal forecasting skill for the High Mountain Asia region in the Goddard Earth Observing System"

General comments:

This manuscript evaluates the seasonal forecast skill for hydrometeorology over the High Mountain Asia (HMA) region from the Goddard Earth Observing System (GEOS). As the author suggests, "S2S forecasting for HMA is in its infancy". Their results show that the GEOS-S2S system's ability to forecast HMA hydrometeorology on the seasonal timescale is limited. The authors raise some issues of the GEOS for seasonal hydrometeorological forecasts. These results help to improve the ability of the seasonal forecast model in the future. Therefore, the scientific questions are of interest. Their introduction provides context and objectives for their work, which catches the reader's interest. The data and methods are described in detail and are reasonable. The results of the evaluation and discussion are well written. In general, this paper is well prepared and fits the scope of ESD.

*Author Comments: We very much thank the reviewer for the time spent on our manuscript and for the generous and kind description of our paper. Our comments below indicate where we made changes in the manuscript to satisfy some of these concerns.*

Specific comments:

1. The credibility of verification data is a great challenge. Even though the authors use multiple data, I still think the current results are quite uncertain due to credibility of verification data.

*Author Comments: We thank the reviewer for this insight. This is the reason for using multiple sources of observed and verification data in our study. Relying on solely one source of information may be misleading, hence we used 2 different products for each climate variable to get a sense of the uncertainty in the forecast skill for each variable. We include a statement in Section 2.3 on Line 182 to state this clearly, "We utilize information from different sources to make sure that evaluation results are not solely dependent on biases or uncertainties in a single reference product. The datasets used for evaluation in our study have their own biases and issues, particularly over the mountainous regions of our study."*

2. The S2S (subseasonal to seasonal) prediction project database (http://www.s2sprediction.net/) provides reforecasts by many operational forecast systems. Can the forecasting skills of the GEOS be compared with models that participate in the S2S prediction project?

*Author Comments:* We agree with the reviewer that this would be an interesting and useful comparison. GEOS-S2S does participate in the North American Multi-Model Ensemble (NMME, https://www.cpc.ncep.noaa.gov/products/NMME/). We provide some literature review in our paper describing the reported skill of the GEOS-S2S system in previous studies, showing GEOS-S2S as a state-of-the-art system. For example, on Line 87 we state:
*"Forecasts are provided to national and international multi-model prediction efforts, including the North American Multi-Model Ensemble (Kirtman et al., 2014). The skill of GEOS-S2S has been reported in various works, such as Gibson et al., (2020) who assessed the hindcast skill of representing ridging events over the Western United States in different S2S models and found the forecast horizon of GEOS-S2S to be comparable with other S2S models in the community."* Furthermore, we also include comments in various parts of the discussion section that compare the skill levels found in this paper with that of S2S prediction skill in other works.

3. It appears that the area of focus includes some low-elevation areas within the range of Figure 1 (e.g., parts of India). Are the authors calculating some statistics (e.g., Figure 2, Table 2) for the entire area of Figure 1? Should low-altitude areas be masked out?

*Author Comments:* The reviewer makes a good point; different regions may have different skill metrics. Our results in Figures 7-12 show maps of the skill and the ensemble spreads for the whole domain, which can be used to analyze spatial differences in the results and to pinpoint specific regions with higher or lower skills in general. Figures such as these can be used to interpret the skill level for different regions within the domain, such as low- or high- elevation regions. For example, Figure 7 shows that there is a higher ensemble spread as well as a higher error for temperature in the India subregion compared to other regions in the domain. To help further satisfy the reviewer's concern, we now provide new figures showing the evaluation metrics for smaller sub-regions within the domain. Figure 1 now shows where these subregions are located, and the revised Figure 3 now shows the evaluation for specific subregions. We describe on Line 326:
*"Figure 3 shows the S2S forecast evaluation based on different subregions within the HMA domain. In Figure 3A-B, … "*

4. Section 3 describes the results in detail. However, there seems to be a lack of an indepth scientific explanation. For example, what are sources and effects of the forecast errors.

*Author Comments:* Section 3 is intended to provide a thorough quantitative analysis of the results. The more in-depth scientific discussion is in Section 4, which digs deeper into the reasoning and qualitative analysis behind the results of Section 3. For example, Section 4.1

discusses the role of model initialization as well as the persistent memory of the physical system and the impact these characteristics have on the skill, and Section 4.3 discusses the role of model resolution or the representation of different processes within the model and how these characteristics impact the skill.

5. I noticed that the skills of GEOS vs. MERRA-2 and observation are quite different (Figure 2a vs. 2b). The ubRMSEs in Figures 5d and 6d show the issue. Which result should I believe? Why are there such obvious differences in skill when using different verification data (especially SM, TWS)? How do the authors interpret such differences of results?

*Author Comments:* We agree. There are distinct spatial and temporal differences in GEOS-S2S skill when compared to MERRA-2 or other observations. There are several reasons that these differences arise for different variables. For example, the fact that GEOS-S2S and MERRA-2 are close in their architecture makes it necessary to have other data for verifying our results, and this is explained in more detail in Sections 4.1 (Lines 495 and 506) and Section 4.3 (Line 585).

We state in Section 4.1 (Line 495) that:
*"When comparing the S2S forecasts with MERRA-2, Figures 2A and 3A show that the snow variables, SM, and TWS have increased skill at early lead time (1-month), and for SM and TWS, this skill can persist for forecasts at longer lead time (2-3 months). This could be because GEOS-S2S and MERRA-2 have similar land conditions during initialization, both modeling systems are quite similar, and because these variables have longer persistence and memory in the physical system. When evaluating the S2S forecasts against MERRA-2, forecast skill is highest in long-memory variables (snow and soil moisture related) and lower in near surface atmospheric variables (T2M and precipitation)."*

We further state (Line 506):
*"Another reason that could explain the skill in certain variables is the role of better land surface initial conditions. For example, fSCA, SWE, SM, and TWS vary more slowly compared to T2M or PRECTOT, and their initial conditions play an important role in the skill of 1-month forecasts. This can be inferred in our results. For example, in Figure 2A the forecast skill relative to MERRA-2 is higher for these variables, perhaps due to similar initialization in the GEOS-S2S and MERRA-2 systems."*

Furthermore, in Section 4.3.3 (Line 585) we mention:
*"Differences in the level of the S2S forecast skill relative to MERRA-2 and to the other reference products (Table 2 and Figure 2) could be due to certain physical processes that are seen in the signatures of the reference data products but under-represented in the frameworks of GEOS-S2S and MERRA-2."*

6. High anomaly correlation or low ubRMSE indicates better forecasting skills. Both the anomaly correlation and ubRMSE represent the correspondence between forecasts and observations. It looks like it is acceptable to use just one metrics. Why use both anomaly correlation and low ubRMSE?

*Author Comments:* Although we agree with the reviewer that the ubRMSE and the anomaly correlation may be showing redundant information in some cases, there are cases in which having both evaluation metrics can be useful. While correlation is a non-dimensional metric that is invariant to changes in the mean and variance, the ubRMSE has units and is sensitive to the variance in the data. In general, most studies that report on a model evaluation show various evaluation metrics. Here, we simply report two metrics that cover complementary skill aspects to make understanding the results of the evaluation more accessible.

7. Section 3.2 and Figure 4: It appears that the annual cycles have large uncertainties, mainly hydrological variables. The anomalies are derived by removing the annual cycle. This might greatly affect the credibility of the results. How does the author address this issue? There should be an explanation.

*Author Comments:* We thank the reviewer for this insightful comment. As a reminder, the anomalies are created by subtracting the mean value for each variable in each respective month (e.g., mean January T2M subtracted from all the January T2M's, mean February T2M subtracted from all the February T2M's, etc.). Section 3.2 shows the annual cycle, which portrays how each variable changes through the course of the year. We see that for some variables, such as T2M and PRECTOT, there is more agreement in the annual cycle between the various products. However there tends to be a higher spread in the annual cycle of the other variables. Some of the reasoning behind this spread are explained in Section 4.3.2. For example, on Line 566, we state:
*"For SM and TWS, error patterns in Figure 5E and 5F and Figure 6E and 6F can primarily be related to monsoon representation in the S2S system, but the errors can also be associated with the observational difference in the seasonal cycles shown in Figures 4E and 4F"*.

8. The reviewer did not get the point of Figure 3. This figure depicts the difference in skill between variables and between forecast lead times. Different variables have different predictability. Forecast skill decreases with forecast lead time as a matter of course. What is the purpose of comparing their relative skills?

*Author Comments:* We thank the reviewer for this comment. This figure was included to visually depict the difference in skill between variables and between forecast lead time. This kind of figures makes it easier to directly compare results between the relative performance skill of the variables/lead times. Furthermore, Figure 2 we show anomaly correlation as the main metric to visualize the differences, whereas Figure 3 uses ubRMSE as the main metric. When a specific box in Figure 3AB is blue, it means that for that variable and at that lead time the skill is higher than that of the skill for that variable at the other lead times, and when a box is red that means the skill is lower for that variable at that lead time compared to the skill for that variable at different lead times. Furthermore, we substituted the older Figure 3 with a new version that includes new analysis that splits up the evaluation into specific subregions, see new Figure 3.

Minor comments:

Line 23 and 25: "ranges" shoud be "range".

*Author Comments:* We thank the reviewer for this comment. We have fixed this in the paper.

Line 118: "…five mountain ranges, including the Himalayas, Inner Tibetan Plateau, Karakoram, and Hindu Kush." Shoud be "four"?

*Author Comments:* We thank the reviewer for this comment. We have fixed this in the paper.

**Final Authors Comments to Reviewer 2**

Comment on egusphere-2022-449
Anonymous Referee #2

Referee comment on "Seasonal forecasting skill for the High Mountain Asia region in the Goddard Earth Observing System" by Elias Charbel Massoud et al., EGUsphere, https://doi.org/10.5194/egusphere-2022-449-RC2, 2022

Summary

This study evaluates the subseasonal predictions from the NASA GEOS5-S2S hindcasts for 1981-2016 over the High Mountain Asia (HMA) domain with a focus on a set of hydrometeorological variables including 2-m air temperature, precipitation, snow cover fraction, snow water equivalent, soil moisture, and total water storage. The evaluation was done against two reanalyses and other independent datasets, and the evaluation focuses on monthly time scale with lead times up to 3 months. Unbiased root mean square error and anomaly correlation are the major metrics used in this evaluation. Overall, the study provides useful information about the predictive skill of the NASA GEOS5-S2S hindcast over the HMA region. The manuscript is well written and easy to understand, and the quality of the visualizations is generally good. However, the study falls short in several important aspect regarding forecast verification at subseasonal to seasonal time scales. The value and the contribution of this study to our understanding about predictability of the climate system at S2S time scale is very limited. I believe at the minimum a major revision is needed. I list my major concerns and some specific comments below.

*Author Comments:* We very much thank the reviewer for the time spent on our manuscript. We believe that the revisions detailed below, including the addition of evaluation for specific sub-regions as well as the addition of new text in various parts of the manuscript, should address the reviewer's concerns.

Major issues

- For prediction beyond the typical weather scale (i.e, 1-2 weeks), probabilistic forecast is more appropriate and useful than deterministic forecast given the chaotic nature of the climate system, which is why S2S forecast with numerical models needs to produce ensemble predictions. In this study, only unbiased root mean square error (ubRMSE) and the anomaly correlation (ACC) of the ensemble mean were used, which is useful but only shows very limited aspects of the forecast quality. There are many verification metrics that can be used for ensemble predictions such as those listed at https://www.cawcr.gov.au/projects/verification/#Methods_for_probabilistic_forecasts. I'd highly recommend that a few more meaningful metrics are included in this study.

*Author Comments:* We thank the reviewer for this comment. We agree that the reliability or the uncertainty in the reported skill of the forecasts is important. For this statement, we point to the ensemble spreads shown in Figures 7-12, which show the spread and therefore the 'reliability' of the forecasts from the model. The higher the ensemble spread in these plots, the less certain the various ensemble members are for each climate variable and lead time. Furthermore, the use of multiple data sources in the evaluation also allows us a look at the uncertainty in our results.

Additionally, we now provide a new section in the discussion (Section 4.2) and new figures showing the comparison between the ensemble spread and the error (Figures S1-S3) with relevant text and references (Section 4.2 Line 521) to provide additional and meaningful probabilistic verification metrics:

*"Other than looking at the forecast error to determine whether a forecast was skillful or not, the spread of the forecast ensemble is another metric that gives indication of reliability when preparing for impacts of weather events. For instance, a smaller spread in the S2S forecasts for a given region might be an indication of higher skill for that variable in that region. The results shown in this study, such as those in Supplementary Figures S1-S3, provide a benchmark of information regarding the forecast skill as well as the ensemble spread in the GEOS-S2S seasonal forecasts. Generally, one can compute the spread/error ratio with the goal of that being close to 1; if it is larger than 1 (more spread than error) this is considered "underconfident", and if it is less than 1 this is considered "overconfident" (Fortin et al., 2014). For the reliability plots in Figures S1-S3, almost all the maps are blue, indicating that the forecasts are overconfident, meaning there is a smaller spread compared to what the error is. However, for SM (Figure S3) this is the opposite, with red indicating that the forecasts are underconfident, meaning there is a larger spread compared to what the error is. Furthermore, for PRECTOT, fSCA, and SWE (Figures S1-S2), there are regions in the Karakoram, Himalayas, and Inner Tibetan Plateau that also show red, indicating that the forecasts are underconfident. Is it important to note, however, that there are limitations to using this reliability metric, including the fact that one can have a "perfect" ensemble prediction system with low correlation between skill and spread (c.f., Hopson 2014), in which case the reliability of the forecasts would be difficult to capture."*

- My biggest concern regarding the analysis is its over-simplified approach to deal with the spatial heterogeneity within the study domain. The study domain is quite large; more importantly, it is very heterogeneous with distinct climates and land surface characteristics including elevation, land cover type, etc. As shown in Figures 7-12, temperature, precipitation and other hydrometeorological variables and model's skill in predicting these quantities can vary drastically across the domain. Spatially averaging them across high mountain ranges, the Tibet Plateau, Taklamakan desert, and the Indian subcontinent does not make much sense, and evaluating the spatially averaged quantities is not very meaningful and insightful. It is not clear what these spatial averages physically mean and how verification at such a level can help us to understand the model deficiency in a meaningful way. Although Figure 7-12 highlight the spatial heteorogeneity, the evaluation is only limited to the ensemble mean, spread, and ubRMSE. I'd suggest that

the authors divide the domain into multiple smaller regions that are more homogeneous or multiple watersheds where the spatial averages are more meaningful, and conduct the forecast verification of these regional quantities using multiple metrics (probabilistic and deterministic).

*Author Comments:* The reviewer makes a good point; different regions may have different forecast skill. Our results in Figures 7-12 show maps of the skill and the ensemble spreads for the whole domain, which can be used to analyze spatial differences in the results and to pinpoint specific regions with higher or lower skills in general. Figures such as these can be used to interpret the skill level for different regions within the domain, such as low- or high-elevation regions. For example, Figure 7 shows that there is a higher ensemble spread as well as a higher error for temperature in the India subregion compared to other regions in the domain. By including the evaluation information spatially, i.e., maps that show ensemble spread and ubRMSE of each grid cell within the whole domain, one can estimate how skillful and reliable the forecasts are for broader regions (such as skill and reliability patterns at the watershed scale) as well as regions that are more local (skill and reliability at individual grid cells). To help further satisfy this concern, we now provide new figures showing the evaluation metrics for smaller sub-regions within the domain. Figure 1 now shows where these subregions are located, and the revised Figure 3 now shows the evaluation for specific subregions. We describe on Line 326:
*"Figure 3 shows the S2S forecast evaluation based on different subregions within the HMA domain. In Figure 3A-B, … "*

Minor issues

- line 13: "where water resource needs change depending on ..." although this sentence is correct, it could be a little confusing as either "needs" or "change" can be interpreted as the verb, thus resulting in different meanings.

*Author Comments:* We thank the reviewer for this comment. We have fixed this in the paper.

- line 13: how is intensity of the hydrological cycle defined? It was not mentioned in the study.

*Author Comments:* We thank the reviewer for this comment. We have fixed this in the paper.

- line 30: "a range of factors", the predictability itself is also an important factor.

*Author Comments:* We thank the reviewer for this comment. We have added this in the paper.

- line 34: remove the comma before "where"

*Author Comments:* We thank the reviewer for this comment. We have fixed this in the paper.

- line 35-36: This sentence reads a little awkward, please consider rephrase.

*Author Comments:* We thank the reviewer for this comment. We have fixed this in the paper.

- line 40: Part of the study domain is heavily populated, but the majority of HMA do not have much population, such as Tibet Plateau and dessert.

*Author Comments:* We thank the reviewer for this comment. We have fixed this in the paper.

- line 43: The term "water tower" of the Earth have been used for many years among researchers in Asia, so some earlier literature needs to be cited here to be more appropriate.

*Author Comments:* We thank the reviewer for this comment. We have added this in the paper.

- line 144: This is only over the real-time forecast period, isn't it? Please clarify that these 6 additional members are not available in the hindcast period and thus not used in the evaluation.

*Author Comments:* We thank the reviewer for this comment. We cleared this up in the paper.

- line 147: "a long period for forecast validation" "validation" and "verification" are different terms although they are related. One can verify if a forecast is correct or wrong, but you cannot validate a forecast when the forecast is wrong. So it would be more appropriate to say "forecast verification" or "forecast evaluation" here.

*Author Comments:* We thank the reviewer for this comment. We have fixed this in the paper.

- line 233: remove "in our evaluation" as it is redundant with "in this study" at the beginning of the sentence.

*Author Comments:* We thank the reviewer for this comment. We removed this in the paper.

- line 234-235: Does this mean the dataset is heterogeneous in space and time? If that is the case, how does this affect the evaluation? Please explain.

*Author Comments:* The product itself is a blended version of multiple different products. So, although the different measurements that are synthesized have varying depths, the output product has a consistent 5cm depth. We have made this more clear in the paper.

- line 282: It would be useful to give the equation for R_anom too. Does this includes both space and time dimensions?

*Author Comments:* Yes, the equation should include both space and time dimensions. We added a new equation to the paper to show this (Eq 2 in the paper).

- line 291-292: Is the ensemble spread also lead-time dependent?

*Author Comments:* Yes, we show in Figures 7-12 that each lead time has a different ensemble spread map. Generally, the ensemble spread increased with lead time for most variables and in most regions, except for precipitation in the Indian Subcontinent (Figure 8F and 8J). We now explain this in the paper on Line 438:
*"Furthermore, the ensemble spread is generally higher in the mountain regions and lower over the Indian subcontinent with increasing lead time (Figure 8F and 8J)."*

- Section 3.2: Since the evaluation metrics are based on anomalies, what purpose does this section serve in the paper?

*Author Comments:* The anomalies are created by subtracting the mean value for each variable in each respective month (e.g., mean January T2M subtracted from all the January T2M's, mean February T2M subtracted from all the February T2M's, etc.). Section 3.2 shows and describes the annual cycle, which portrays how each variable changes through the course of the year.

- Figure 4 and others: Since the gridded model forecast is spatially averaged over the large domain with different masks for different variables, it would be useful to show the masks for these variables in Figure 1 so that readers know how the spatial average is calculated.

*Author Comments:* We thank the reviewer for this comment. These masks are shown in Figures 9 for fSCA, Figure 10 for SWE, and Figure 11 for SM. Grid cells that are white are the ones that are masked out. This is also mentioned in the captions of Figure 9-11 and in the paper on Lines 450 and 479.

- Section 3.3: This section is about the absolute error. Because of the seasonality discussed in the previous section, it is not surprising that errors are generally larger during the season when the absolute value of variable is also large. So it will be necessary and more informative to discuss the relative errors beyond the absolute error.

*Author Comments:* We thank the reviewer for this comment. Note, however, that the seasonality of the errors is not totally dependent on the absolute value of the variable. For example, Figures 5A and 6A show that error for T2M is higher in the winter months, when T2M is in fact lower. These types of figures are useful to show this seasonality in the forecast error for each variable and how they differ between the different evaluation products. Furthermore, the ubRMSE calculates the error based on the anomaly forecasts, and so in that sense the measure of error is a relative error since it is based on anomalies and not absolute values.

- Figure 6: For each panel,the y-axis should be set to the same range as that in the

corresponding panel in Figure 5.

*Author Comments:* We thank the reviewer for this comment. This might be helpful if the figures were being used solely to compare the errors based on the evaluation products, but the main point here is to show the seasonality of the errors for each variable. Since there is a large range for some variables between Figures 5 and 6 (e.g., for TWS), we do not believe it would be useful to make the y-axis the same in these figures, because this would make it difficult to visually notice the seasonality in some plots. We now have a statement in the paper to explain the difference in the y-axis for Figures 5 and 6 on Line 379:
*"Note, the y-axes in Figures 5 and 6 are different so that the seasonality in each figure is properly portrayed."*

- Line 489-490: The results in this study do not seem to back up this statement.

*Author Comments:* We thank the reviewer for this comment. The results that support this statement are shown in Figures 2A and 2B and explained in the paragraph starting on Line 516. To make the explanation not sound too speculative, we have changed some of the wording in this sentence:
*"Therefore, forecast skill in shorter memory variables (T2M, PRECTOT) may increase with improvements in resolution and process representation, and gains in forecast skill for longer memory variables (fSCA, SWE, SM, and TWS) may be achieved with improved land surface initial conditions, and if successful, increased forecast skill in 1-month lead time can propagate through to longer leads."*

- Line 518-520: This statement is speculative. It would be more appropriate to provide justifications.

*Author Comments:* We thank the reviewer for this comment. The results that support this statement are shown in Figures 4E and 4F and explained in Section 3.2. As for the role of the monsoon, the statement might have sounded over-confident, therefore we changed some wording in the sentence on Line 566:
*"For SM and TWS, error patterns in Figure 5EF and Figure 6EF may be related to monsoon representation in the S2S system, but the errors can also be associated with the observational difference in the seasonal cycles shown in Figures 4E and 4F."*

- Line 529: How is 4% cold bias calculated? Using different units such as Kelvin, Celsius will certainly result in different percentage change? So a statement like this does not make much sense.

*Author Comments:* We thank the reviewer for this comment. These results are reported from a different paper (Hsu et al., 2021). We have edited these comments to remove any text that can be confusing.

- Line 603-604: This statement assumes that the ensemble spread of the forecast is

informative. The assumption may or may not be true. Linking a smaller forecast spread with higher skill is unjustified and questionable.

*Author Comments:* We thank the reviewer for this comment. We mention that this 'might' be an indication of more reliability, which is true in many cases. We now provide a new section in the discussion (Section 4.2) and new figures (Figures S1-S3) showing the comparison between the ensemble spread and the error. Generally, one can compute the spread/error ratio with the goal of that being close to 1 (Fortin et al., 2014). As mentioned above, this is explained in detail in Section 4.2 on Line 520.

- Line 633-634: It is not clear how this study achieve this as it does not provide much insights that can guide model improvements.

*Author Comments:* We thank the reviewer for this comment. Although we believe that the results and corresponding conclusions are explained in the text (e.g., TWS having high errors compared to GRACE since the model does not have groundwater pumping whereas the GRACE signature includes this process), we now add more details in this section of the paper on Line 685 to avoid any confusion:
*"These improvements can help, for example, with forecasts of TWS since the model does not have groundwater pumping whereas the GRACE signature includes this process."*

Comment on egusphere-2022-449
Anonymous Referee #3

Referee comment on "Seasonal forecasting skill for the High Mountain Asia region in the Goddard Earth Observing System" by Elias Charbel Massoud et al., EGUsphere, https://doi.org/10.5194/egusphere-2022-449-RC3, 2022

This paper examines the seasonal prediction skill of the NASA's Goddard Earth Observing System (GEOS) S2S prediction system over High Mountain Asia for a series of hydrological variables. A set of observational data and the MERRA-2 and ERA-5 re-analyses are used as benchmarks.

Monthly means of reforecasts over the period 1981 to 2016 are analysed.

The paper provides a clear and well written description of the system and presents a clear analysis the forecasts of these hydrological variables. It provides a solid detailed reference to further assess more specific aspects of the performance of the GEOS S2S prediction system over HMA. The Figures are clear but, while many detailed descriptions are provided, I find the paper a little thin on new science and on understanding potential sources of predictability. At the very least, some more discussion points should be included.

*Author Comments:* We very much thank the reviewer for the time spent on our manuscript. We have included more discussion points that address this concern in the new version of the paper, including the addition of evaluation for specific sub-regions, plots showing reliability metrics, as well as the addition of new text in various parts of the manuscript.

MAIN COMMENTS

1) The skills over this large, heterogenous region are small, especially when measured against independent data. I wonder if such small skills are relevant at all. The prediction system may have higher skill in some variables, in more limited domains or at certain times of year.

Low skill is not unexpected given the large area, the varied topography (from lowlands to high mountains) and land cover, and the different regional climates of this large region. For example, only a small Southeastern part of the Tibetan Plateau (and hence, only parts of the HMA) is influenced by the Indian summer monsoon (ISM).

It is well-known that the skills depend on the verification data, and they would be higher, in most cases, when verified MERRA-2 given the parent model. It would be of interest to verify some variables against several datasets, or better, against merged datasets that

take into account uncertainties in the various observations. I realise that such merged dataset might not exist over this region, but the point could be mentioned in the Discussion.

*Author Comments:* We thank the reviewer for this insightful comment. We do understand how the skills reported here can seem minor and not relevant, however we would like to point out that the skills shown are based on anomaly skill (e.g., anomaly correlation and unbiased RMSE). Therefore, any skill that is above 0 anomaly correlation is relevant, since it essentially means that it is more skillful than just guessing the mean climate. For S2S predictions at a range of 1-3 months, we believe that this is rather useful and promising.

As for gaining results in the more limited domains, our results in Figures 7-12 show maps of the skill and the ensemble spreads for the whole domain, which can be used to analyze spatial differences in the results and to pinpoint specific regions with higher or lower skills in general. Figures such as these can be used to interpret the skill level for different regions within the domain, such as low- or high- elevation regions. For example, Figure 7 shows that there is a higher ensemble spread as well as a higher error for temperature in the India subregion compared to other regions in the domain. To help further satisfy this concern, we now provide new figures showing the evaluation metrics for smaller sub-regions within the domain. Figure 1 now shows where these subregions are located, and the revised Figure 3 now shows the evaluation for specific subregions. We describe in the paragraph starting on Line 326: "*Figure 3 shows the S2S forecast evaluation based on different subregions within the HMA domain. In Figure 3A-B, … "*

Regarding the use of multiple data sets or perhaps a merged data product, this is the reason we use multiple sources of observed and verification data in our study. Relying on solely one source of information may be misleading, hence we used 2 different products for each climate variable to get a sense of the uncertainty in the forecast skill for each variable. We are not aware of a merged data product for this region, perhaps the MERRA-2 product is a close example of what the reviewer is describing here. In the discussion, we have added statements on Line 652 about the potential of having a data set like this and how it can be helpful for the evaluation:
"*We are not aware of a merged data product for the HMA region, which would be extremely valuable for an evaluation study like this, but perhaps the combination of MERRA-2 data with other verification data products is a good alternative.*"

2)Concerning actual societal needs, the reliability of such forecasts is a question of utmost importance that needs to be addressed in a probabilistic context. Is it possible to quantify the reliability of the forecasts with the current system using standard metrics? At least, the outlook could be mentioned in the Discussion.

*Author Comments:* We thank the reviewer for this comment, and we agree that the reliability or the uncertainty in the reported skill is important especially for societal needs. For this statement, we aim to address forecast reliability in a probabilistic sense by comparing

ensemble spreads vs. errors. We point to the ensemble spreads shown in Figures 7-12, which show the spread and therefore the 'reliability' of the forecasts from the model. The higher the ensemble spread in these plots, the less certain the various ensemble members are for each climate variable and lead time. Furthermore, the use of multiple data sources in the evaluation also allows us a look at the uncertainty in our results. We have added statements on Line 649 in the discussion to elaborate on the interpretation of uncertainty and how it impacts societal needs:

*"For this study, we use multiple sources of observed and verification data to estimate the forecast skill since relying on solely one source of information may be misleading. Here, we used two different products for each climate variable to get a sense of the uncertainty in the forecast skill for each variable."*

We have also provided additional material to help describe the reliability of the forecasts and how that can be estimated using standard reliability metrics. For example, we now provide new a new section in the discussion (Section 4.2) and new figures showing the comparison between the ensemble spread and the error (Figures S1-S3) with relevant text and references (Line 521) to provide additional and meaningful verification metrics:

*"Other than looking at the forecast error to determine whether a forecast was skillful or not, the spread of the forecast ensemble is another metric that gives indication of reliability when preparing for impacts of weather events. For instance, a smaller spread in the S2S forecasts for a given region might be an indication of higher skill for that variable in that region. The results shown in this study, such as those in Supplementary Figures S1-S3, provide a benchmark of information regarding the forecast skill as well as the ensemble spread in the GEOS-S2S seasonal forecasts. Generally, one can compute the spread/error ratio with the goal of that being close to 1; if it is larger than 1 (more spread than error) this is considered "underconfident", and if it is less than 1 this is considered "overconfident" (Fortin et al., 2014). For the reliability plots in Figures S1-S3, almost all the maps are blue, indicating that the forecasts are overconfident, meaning there is a smaller spread compared to what the error is. However, for SM (Figure S3) this is the opposite, with red indicating that the forecasts are underconfident, meaning there is a larger spread compared to what the error is. Furthermore, for PRECTOT, fSCA, and SWE (Figures S1-S2), there are regions in the Karakoram, Himalayas, and Inner Tibetan Plateau that also show red, indicating that the forecasts are underconfident."*

3)I wonder about the relationship between surface temperature and the snowpack. Is there is a strong coupling between the two in the forecasts during some months? This could provide a source of skill.

*Author Comments:* We thank the reviewer for this insightful comment. We do not find any direct relationship between the skill in temperature and how it affects the skill in the snowpack. This is an idea that deserves some discussion, and we now provide some additional details in the paper on Line 604 to discuss this:

*"Appropriate representation of seasonal snow along with temperatures and antecedent precipitation are critical to realistically forecasting the HMA energy and water cycles. GEOS-S2S forecasts tend to underestimate temperature and overestimate precipitation relative to both*

*MERRA-2 and the reference observations during all months and nearly all lead times (Figure 4A-B); this cumulatively impacts snow cover and volume (Figure 4C-D)."*

4) Improved prediction of the circulation could lead to improved skill. The authors mention the importance of the ISM. I believe that wintertime precipitation over the northern part of HMA is brought by the so-called westerly disturbances. The authors could mention in the Discussion, whether the dynamics and the associated with precipitation is well represented in the forecast.

*Author Comments: We again thank the reviewer for this insightful comment. We do report on the importance of the ISM in Section 4.3.2. We also now include some details on Line 562 about westerly disturbances and how the representation of that process within the model can impact the forecast skill:*
*"Another explanation for this could be the role of westerly disturbances, which bring enhanced precipitation during the winter months for the west and northern parts of HMA (Cannon et al., 2016), where in our analysis the precipitation forecasts for these regions are underconfident (Figure S1) and larger errors for fSCA (Figure 9CD) and SWE (Figure 10CD) can be expected in these regions."*

5)There has been a significant effort in recent years to assess the impact of land initialisation (esp. snow, soil moisture) in S2S and seasonal forecasts and some studies are relevant for the HMA region yet there is little mention of that relevant literature.

Koster, R. D., Mahanama, S. P. P., Yamada, T. J., Balsamo, G., Berg, A. A., Boisserie, M., et al. (2011). GLACE2: The second phase of the global land atmosphere coupling experiment: Soil moisture contribution to subseasonal forecast skill. Journal of Hydrometeorology,12(5), 805–822.

Senan, R., Orsolini, Y.J., Weisheimer, A. et al. Impact of springtime Himalayan–Tibetan Plateau snowpack on the onset of the Indian summer monsoon in coupled seasonal forecasts. Clim Dyn 47, 2709–2725 (2016). https://doi.org/10.1007/s00382-016-2993-y

*Author Comments: We thank the reviewer for providing these works in the literature. We do report on the impact of land initialization in Section 4.1, and we now also include the references above in the new version of the paper on Line 660.*

MINOR COMMENTS

- The words Seasonal forecasts and S2S forecasts seem to be used loosely throughout the paper. The seasonal forecasts are 9-month long but only the first 3 months are analysed. Some operational centers have different set-ups for Seasonal and S2S prediction systems. The authors could double check that S2S is used as it is meant.

*Author Comments:* We thank the reviewer for this comment. From further discussion, we have determined that the exact difference between S2S vs seasonal forecasting is still difficult to define. Therefore, for the purposes of our paper, we have changed all the text that mentions 'seasonal forecasting' and we now consistently say 'S2S forecasting. We have checked that our paper is consistent in the terminology throughout, with the hopes of eliminating any confusion.

- It was not clear to me whether total precipitation is liquid precipitation or if it contains also solid precipitation.

*Author Comments:* The form of precipitation analyzed in this paper is the PRECTOTCORR variable from the GEOS-S2S system, and is derived from the PRECTOT variable, which is total precipitation including rain and snow, i.e., PRECTOT = liquid + solid (total) precipitation. We now include a comment in the paper to better explain this on Line 162.

- The information on ensemble size should be presented more clearly (Abstract, or Table)

*Author Comments:* We thank the reviewer for this comment. We now provide a better and more clear description of the ensemble size in Section 2.2. On Line 148 we state:
*"All forecasts are 9 months in duration, but, we focus here on the 4 ensemble members in the retrospective forecasts with 1-, 2-, and 3-month lead times."*

Wording

L58: the foothills of the Himalayas perhaps better than the foot of the Himalayas (?)

*Author Comments:* We thank the reviewer for this comment. We have fixed this in the paper.

L560: precipitation is used twice in same sentence.

*Author Comments:* We thank the reviewer for this comment. We have fixed this in the paper.